# Glyco-Decipher enables glycan database-independent peptide matching and in-depth characterization of site-specific N-glycosylation

Zheng Fang [1,2,5], Hongqiang Qin[1,2,5], Jiawei Mao[1], Zhongyu Wang[1,2], Na Zhang[1], Yan Wang[1], Luyao Liu[1,2], Yongzhan Nie[3], Mingming Dong [1,4✉] & Mingliang Ye [1,2✉]

Glycopeptides with unusual glycans or poor peptide backbone fragmentation in tandem mass spectrometry are unaccounted for in typical site-specific glycoproteomics analysis and thus remain unidentified. Here, we develop a glycoproteomics tool, Glyco-Decipher, to address these issues. Glyco-Decipher conducts glycan database-independent peptide matching and exploits the fragmentation pattern of shared peptide backbones in glycopeptides to improve the spectrum interpretation. We benchmark Glyco-Decipher on several large-scale datasets, demonstrating that it identifies more peptide-spectrum matches than Byonic, MSFragger-Glyco, StrucGP and pGlyco 3.0, with a 33.5%-178.5% increase in the number of identified glycopeptide spectra. The database-independent and unbiased profiling of attached glycans enables the discovery of 164 modified glycans in mouse tissues, including glycans with chemical or biological modifications. By enabling in-depth characterization of site-specific protein glycosylation, Glyco-Decipher is a promising tool for advancing glycoproteomics analysis in biological research.

[1] CAS Key Laboratory of Separation Science for Analytical Chemistry, Dalian Institute of Chemical Physics, Chinese Academy of Sciences, 116023 Dalian, China. [2] University of Chinese Academy of Sciences, 100049 Beijing, China. [3] State Key Laboratory of Cancer Biology, National Clinical Research Center for Digestive Diseases and Xijing Hospital of Digestive Diseases, Fourth Military Medical University, 710032 Xi'an, China. [4] School of Bioengineering, Dalian University of Technology, 116024 Dalian, China. [5] These authors contributed equally: Zheng Fang, Hongqiang Qin. ✉email: dongmm@dlut.edu.cn; mingliang@dicp.ac.cn

N-linked glycosylation is a common and heterogeneous post-translational modification (PTM) of proteins that mediates a wide range of biological processes, including protein folding[1], immunosuppression activity[2], and cancer progression[3]. The high macro- and micro-heterogeneity of protein glycosylation make the analysis of glycosylation extremely challenging[4]. Mass spectrometry (MS)-based analysis of intact glycopeptides has been shown to be indispensable in the characterization of site-specific glycans at the proteome level in functional studies[5–7]. In the past decade, many glycan database-dependent tools have been developed to analyze the glycoproteomics data acquired via tandem mass spectrometry (MS/MS)[8–13] and their performance is closely associated with the adopted search variables including the glycan database[14]. The immense diversity of glycans largely increases computational complexity in the identification of glycopeptides. Proteome-wide N-glycan databases, for example, the GlyTouCan database[15], contain over 1500 unique monosaccharide compositions with mass values ranging from hundreds to over 6,000 Daltons (Da) (Supplementary Data 1). The numerous and diverse types of glycans necessitate an extended search space for glycopeptide identification when compared with other protein modifications. In addition, the fragmentation of peptide backbones is typically suppressed in higher-energy collisional dissociation (HCD)-MS/MS analysis due to the labile nature of glycosidic bonds, which hinders the identification of glycopeptides. For example, in a recently published glycoproteomics dataset of mouse tissues, only 6% of the collected glycopeptide spectra were reliably annotated in the original study[9] (Supplementary Table 1), a rate that lags far behind the routine proteomic analysis, for which more than 70% of spectrum identification rate has been reported[16].

Most of the developed glycoproteomics tools rely on glycan databases to interpret glycopeptide spectra. A straightforward search strategy, adopted by Byonic[8] and SEQUEST[17], is to set the glycans in a database as variable modifications on peptides for proteome database searching, which was shown to be of limited identification sensitivity[10]. An alternative approach is to iteratively match the glycan masses to the glycopeptide spectra to identify the peptide backbones[18]. This method, called mass offset search, decreases the search space for peptide identification and has been adopted in several cutting-edge tools, including MSFragger-Glyco[10] and pGlyco 3.0[13]. These glycan database-dependent tools have substantially improved glycoproteomics analysis but fail to identify glycopeptides with unexpected glycans. In addition to the differences in monosaccharide composition, the diversity of glycans is further increased by the glycan modifications introduced in vivo/in vitro after glycan biosynthesis[19]. Modifications on glycans have been implicated in the important biological functions of glycoproteins[20,21]. The vast diversity of post-translational and chemical modifications on peptides have been systematically investigated by using the open search method[16,22–24], yet there is still no method for discovering modifications on glycans at the glycoproteomics level.

Here, we present a platform called Glyco-Decipher to interpret the glycoproteomics data of N-linked glycopeptides (Fig. 1). It adopts a glycan database-independent peptide matching scheme that allows the unbiased profiling of glycans and the discovery of unexpected glycans linked with modifications. The fragmentation patterns of peptide backbones, which are highly similar for glycopeptides with identical backbones, are exploited to improve the spectrum interpretation. Reanalysis of several large-scale datasets showed that Glyco-Decipher outperformed other existing software tools in glycopeptide identification. Our glycan database-independent search also provides a more comprehensive insight into the site-specific glycosylation and enabled the unveiling of diverse modification moieties on glycans along with as many as

164 modified glycans beyond the GlyTouCan database entries. We expect that Glyco-Decipher will become a versatile tool for studying site-specific N-glycosylation and will benefit the glycobiology field.

## Results

**Development of Glyco-Decipher.** Glyco-Decipher begins with in silico deglycosylation[25,26] to identify peptide backbones in glycopeptides without setting any glycan modifications (Fig. 1). The core structure of N-glycan on glycopeptides generates a series of Y fragment ions (i.e., Y0, Y-HexNAc(1), Y-HexNAc(2), Y-Hex(1)HexNAc(2), Y-Hex(2)HexNAc(2), and Y- Hex(3)HexNAc(2)) with known mass gaps during HCD fragmentation (Fig. 2a). These Y fragment ions in the glycopeptide spectra are determined by matching mass gaps to derive the masses of peptide backbones. The in silico deglycosylated spectra are generated by the removal of B/Y ions originating from glycan fragmentation followed by the reassignment of the precursors to peptides ("Methods"). MS-GF + [27], which is an open-source database search tool integrated in Glyco-Decipher, is then used to search for the resulting spectra in proteome databases for the initial identification of peptide backbones. For a second-stage search, the fragmentation patterns of each peptide backbone are extracted from the peptide-spectrum matches (PSMs) and applied to match glycopeptide spectra that remained unassigned in the initial search. This strategy, called spectrum expansion, exploits the high similarity of peptide fragmentation patterns for glycopeptides with identical backbones and enables the identification of glycopeptide spectra with poor peptide fragmentation. No glycan information is used in either stage of peptide identification. After the identification of peptide backbones, the mass of the glycan part is precisely derived by the mass difference between the precursor and the peptide backbone. This glycan database-independent peptide identification pipeline enables Glyco-Decipher to profile glycans at the proteome level and to discover glycans not present in databases.

To identify the attached glycans, candidate glycans in the built-in (GlyTouCan) or user-provided database are scored by matching their theoretical fragment ions with experimental B/Y ions in glycopeptide spectra (Supplementary Figs. 1–3). For glycans that do not match any database entries, Glyco-Decipher performs monosaccharide stepping to reveal the composition of the glycans and their potential modifications. Target-decoy spectrum matching and expectation value calculation[28] are adopted to evaluate the false discovery rate (FDR) at both identification stages of peptides and glycans (Supplementary Fig. 4). Glycans attached to peptide backbones are mapped to the site level for further analysis. A quantification module that extracts the elution profiles of intact glycopeptides in MS1 is embedded in Glyco-Decipher for abundance analysis of protein glycosylation.

**Utilization of peptide fragmentation patterns to improve spectrum interpretation.** A dataset containing 1,386,844 tandem mass spectra acquired by stepped collision energy HCD (sceHCD) fragmentation for intact glycopeptides from five mouse tissues[9] (brain, heart, kidney, liver, and lung) was used for the performance evaluation. From the 1,282,263 oxonium ion-containing glycopeptide spectra, Glyco-Decipher yielded a total of 140,250 peptide-spectrum matches by the in silico deglycosylation scheme (Supplementary Note 1 and Supplementary Figs. 45–48). Notably, the in silico deglycosylation scheme only permits the identification of peptide backbones for the glycopeptide spectra with sufficient peptide fragment ions. We then proposed that the peptide fragmentation patterns in these identified spectra could be used to facilitate the interpretation of

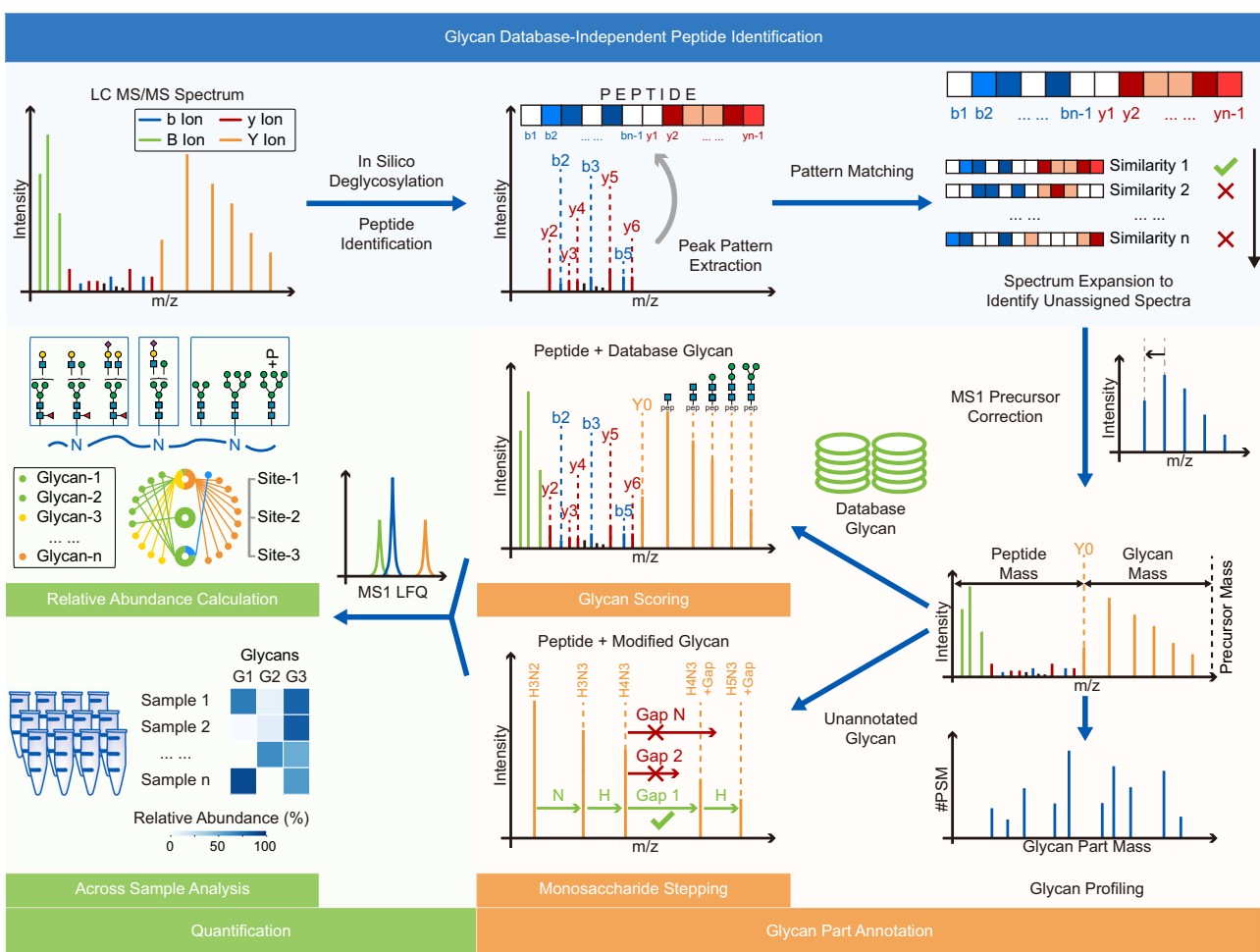

**Fig. 1 Workflow of Glyco-Decipher.** Glyco-Decipher contains three modules: (1) Glycan database-independent peptide matching. The in silico deglycosylated spectra are searched against the protein database without setting any glycans as modifications, which determines the peptide backbone for glycopeptide spectra with rich peptide fragment ions. Then, the fragmentation patterns of the peptide backbones are extracted and utilized to match spectra that remain unannotated. This step, termed "spectrum expansion", enables the identification of peptide backbones of glycopeptide spectra with poor peptide fragmentation. (2) Glycan annotation. The mass of the glycan part is precisely derived by the mass difference between the precursor and the peptide backbone. The mass profile of glycans in a system is constructed without the use of glycan databases. For glycan annotation, the experimental B/Y ions in glycopeptide spectra are first matched to their theoretical fragment ions of database glycans to identify glycans. For glycans that do not match any database entries, Glyco-Decipher performs monosaccharide stepping to reveal the composition of modified glycans and potential modification moiety on them. (3) Quantification. The quantification module based on the elution profiles of glycopeptides is embedded in Glyco-Decipher and allows the computation of the abundance distributions of site-specific glycans.

spectra with poor peptide fragmentation. To this end, we first evaluated the similarity of peptide fragmentation patterns in the spectra of glycopeptides sharing the same backbones. Figure 2b presents fragmentation patterns of the peptide backbone "MHLNGSNVQVLHR" modified by diverse glycans in the six glycopeptide spectra identified by the in silico deglycosylation scheme from a single liquid chromatography (LC)-MS/MS analysis of mouse liver samples. Their patterns are highly similar: the b2 ion is always the most abundant peptide fragment ion, followed by the b3 and other ions. We then investigated the fragmentation patterns of the peptide backbones in the identified glycopeptide spectra of the five mouse tissue datasets. Among the fragmentation patterns in the 140,250 PSMs achieved by in silico deglycosylation, the majority (127,095, 90.6%) exhibited over 0.9 cosine similarities with the average pattern of the corresponding peptide backbone (Fig. 2c), regardless of the attached glycans and/or precursor charge states (Supplementary Fig. 5). This result confirmed that N-glycopeptides sharing the same peptide backbone exhibit similar peptide fragmentation patterns.

We then applied the obtained peptide fragmentation patterns to improve the interpretation of glycopeptide spectra (Methods) and termed this process "spectrum expansion". Figure 2d illustrates an example of spectrum expansion: the peptide backbone "MHLNGSNVQVLHR" was initially identified by in silico deglycosylation of six glycopeptide spectra (green, Fig. 2d), and the averaged fragmentation pattern was used to match the unassigned spectra in spectrum expansion to achieve additional PSMs (blue). The results demonstrated that high-score PSMs were observed within three narrow retention time windows at 70, 80, and 92 min, respectively (note that an LC gradient of as long as 360 min was adopted for glycopeptide separation in this dataset). While random matches (orange) and low-score PSMs spanned the whole chromatography gradient. From the final glycopeptide identification results (Supplementary Table 3), we found that those made in the in silico deglycosylation stage (green, approximately 70 min) corresponded mainly to glycopeptides with oligo-mannose glycans, while those from spectrum expansion were glycopeptides with glycans containing one

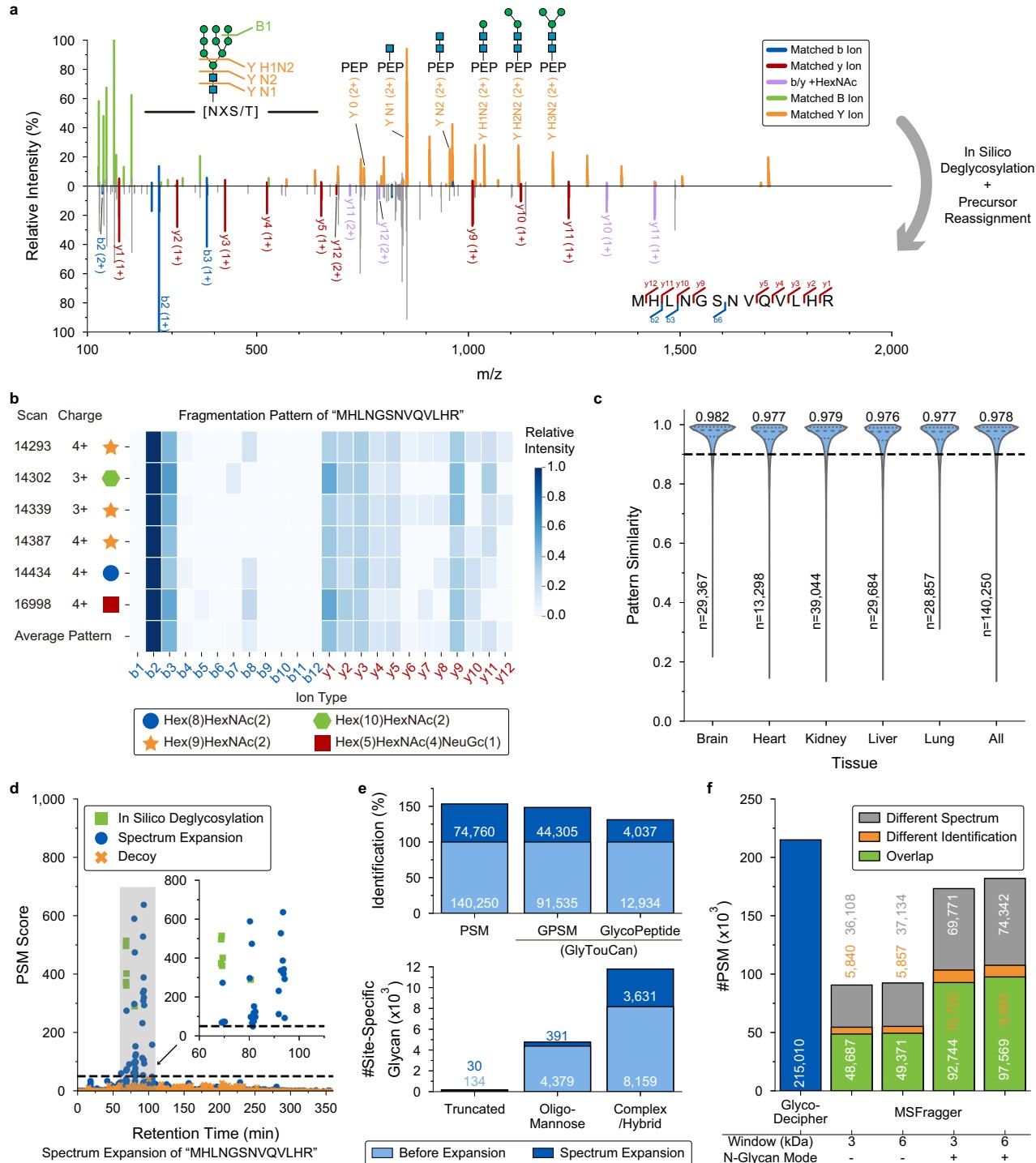

(~80 min) or two (~92 min) sialic acids. Comparative analysis of their spectra indicates that fewer and lower-intensity peptide fragment ions were generated in the spectra of sialic acid glycopeptides than in the spectra of oligo-mannose glycopeptides (Supplementary Fig. 6). However, the consistent fragmentation pattern of the peptides elevated the matching scores of true-positive PSMs and thereby enabled the interpretation of these glycopeptide spectra. More examples of peptide identification in spectrum expansion are shown in Supplementary Figs. 7–9.

By employing the spectrum expansion strategy (Supplementary Fig. 10), peptide backbones in 53.3% (74,760/140,250) more spectra were uncovered and resulted in 215,010 PSMs without the

use of glycan databases (Fig. 2e, top). The glycans in glycopeptide spectra were matched to the downloaded GlyTouCan database, which contains 1,766 unique monosaccharide compositions (Methods). In total, a 48.4% increase (before: 91,535, after: 135,840) in the number of glycopeptide-spectrum matches (GPSMs, PSMs with annotated glycan compositions) and a 31.2% increase (before: 12,934, after: 16,971) in the identification of unique glycopeptides were achieved after spectrum expansion (Fig. 2e, top, Supplementary Data 2). Substantial site-specific glycans of complex/hybrid type were revealed at the expansion stage, resulting an improvement of 44.5% in the number of identification results after expansion (Fig. 2e, bottom). Statistical

**Fig. 2 Peptide fragmentation patterns improve spectrum interpretation and glycopeptide identification. a** Tandem mass spectrum example of glycopeptide "MHLNGSNVQVLHRLTIR- Hex(9)HexNAc(2)" (top) and the in silico deglycosylation result of the spectrum (bottom). Blue: b ions of peptide; red: y ions of peptide; purple: b/y ions with HexNAc residue; green: B ions; orange: Y ions of glycan with intact peptide backbone attached. **b** Fragmentation patterns of the peptide backbone "MHLNGSNVQVLHR" modified by different glycans and/or with different precursor charge states in six glycopeptide spectra. **c** Distributions of similarities between the peptide fragmentation pattern in each peptide-spectrum match (PSM) from in silico deglycosylation and the averaged pattern of the corresponding peptide backbone. The quartiles of the distributions are indicated by inner dashed lines. The medians and the spectrum numbers are labeled in the plot. The similarity value of 0.9 is indicated by an outer dashed line. Source data are provided as a Source Data file. **d** Distribution of matching scores for the peptide backbone "MHLNGSNVQVLHR" with glycopeptide spectra across the entire data acquisition time window. Specifically note that the score of PSM from in silico deglycosylation was re-calculated with the consideration of peptide fragmentation pattern. Inset: The score distribution of PSMs after score filtration and core structure peak matching in spectrum expansion. The dashed line indicates PSM score threshold of 52.75 derived from the e-value filtration method. Green box: target PSMs obtained by in silico deglycosylation; blue circle: target PSMs obtained in spectrum expansion; orange cross: decoy PSMs generated in spectrum expansion for quality control. **e** Comparison of identifications before (pale blue) and after (blue) spectrum expansion. The glycan part was matched with the GlyTouCan database to obtain glycopeptide-spectrum match (GPSM, PSM with definite glycan composition) identifications. Site-specific glycans were classified into three categories based on the glycan composition: truncated glycans (Hex(<4)HexNAc(<3)Fuc(<2)), oligo-mannose glycans (Hex(>3)HexNAc(2)Fuc(<2)) and complex/hybrid glycans. **f** Performance comparison between Glyco-Decipher and MSFragger (V3.1.1) in open search mode. Green: the consistently identified spectra. Orange: the spectra commonly identified, but matched to different peptides. Gray: spectra specifically matched by MSFragger.

analysis also confirmed that the spectrum expansion strategy can be used to determine peptide backbones for glycopeptide spectra with poor peptide fragmentation (Supplementary Fig. 11). Notably, the peptide fragmentation patterns are nearly unchanged in the expansion results (Supplementary Fig. 12), indicating the high confidence of spectrum expansion. Overall, nearly 53.3% more glycopeptide spectra and 31.2% more unique glycopeptides were uncovered in the five mouse tissues after spectrum expansion.

**Comparison with open search method in blind glycan modification search.** Based on the in silico deglycosylation and the subsequent spectrum expansion, the peptide backbones in glycopeptides were confidently identified. In addition, the mass difference between the precursor and peptide backbone was recognized as the attached glycan and the distribution of delta mass values, i.e., the glycan profile for a specific sample could be obtained without the use of glycan databases. As noted above, 215,010 PSMs were identified from the glycopeptide spectra of five mouse tissues. After derivation of glycan masses, distinct glycan profiles were observed across these tissues and the determined glycan masses were located mainly in the range of 1200–4000 Da (Supplementary Fig. 13). The glycan profiles of the mouse tissues led to the observation of 580 distinct mass bins (>10 PSMs for each) in which only 313 were matched to Gly-TouCan database entries, indicating the high diversity of glycans and the presence of unexpected glycans. An alternative approach for profiling N-glycans is to analyze the enzymatically released glycans, from which information on glycosites and glycoproteins is missing. Relevant information is retained in the glycopeptide spectra and is provided by Glyco-Decipher with the glycan database-independent pipeline.

Peptide matching in modification-blind mode could also be achieved by the open search method, which is an effective way to discover unknown modifications on a proteome scale[16,22–24]. The underlying principle of this method is matching the spectrum with all peptide candidates within a wide window around the precursor mass, and the mass difference between the precursor and the peptide sequence is attributed to the detected modification. The open search method was also applied in recent glycoproteomics studies. A total of 300 O-glycan mass values were identified from human kidney tissue, serum, and T cells by using MSFragger in open search mode[10], and the glycan diversity within bacterial proteomes was investigated by wide-tolerance (up to 2000 Da) open searching[29].

To compare the performance between Glyco-Decipher and the open search method in blind glycan searching, we downloaded MSFragger (v3.1.1) and performed an open search with various parameters (Fig. 2f, mass window: 3000/6000 Da, labile search for N-glycan: off/on). Compared to the traditional open search and the N-glycan mode open search of MSFragger, Glyco-Decipher offered over onefold and ~20% increases in the number of PSMs, respectively (Fig. 2f). Approximately 60% of glycopeptide spectra identified by the open search were commonly matched by Glyco-Decipher, and ~90% of these common spectra were matched to identical peptide sequences. The theoretical Y ions of N-glycan core structure were deduced based on the peptide results and were matched in a much higher proportion of PSMs identified by Glyco-Decipher than that by MSFragger (Supplementary Fig. 14a). The Y1 (peptide + HexNAc) ion, which is a general feature of glycopeptide spectra in the dataset acquired upon optimal fragmentation energy[9,30], was matched in all the PSMs of Glyco-Decipher since corresponding core structure ion screening was performed. While only 50% (traditional mode) or 60% (N-glycan mode) of the PSMs specifically identified by MSFragger matched corresponding Y1 ions (Supplementary Fig. 14). Further analysis on the spectra that were inconsistently identified in Glyco-Decipher and MSFragger suggests that the lack of corresponding core structure ions in MSFragger may originate from false-positive peptide identifications in open search (Supplementary Fig. 15). Dissimilar to routine modifications attached to peptides, N-glycans hold a much wider mass distribution, which greatly exceeds the mass window of 500 Da that is typically adopted in an open search[22,23]. An enlarged window (3000/6000 Da in this study) is needed in an open search to cover the mass of N-glycans, leading to a substantial increase in the peptide search space, which results in limited identification sensitivity and highly random hits due to the interference of glycan ions. In stark contrast, a mass tolerance of several ppm was adopted in Glyco-Decipher for peptide close searching without setting any glycan modifications, which requires several orders of magnitude lower search space than the open search.

**Monosaccharide stepping enables the discovery of modified glycans at the proteome level.** After profiling the glycans in mouse tissues, we compared the derived mass values (Supplementary Fig. 13) with GlyTouCan database values. The glycan masses of many PSMs with confident peptide matching did not match any database glycans. For example, from the mouse liver dataset, we identified peptide sequences for 44,646 glycopeptide spectra. Glycans in 27,855 (62.4%) PSMs were annotated by

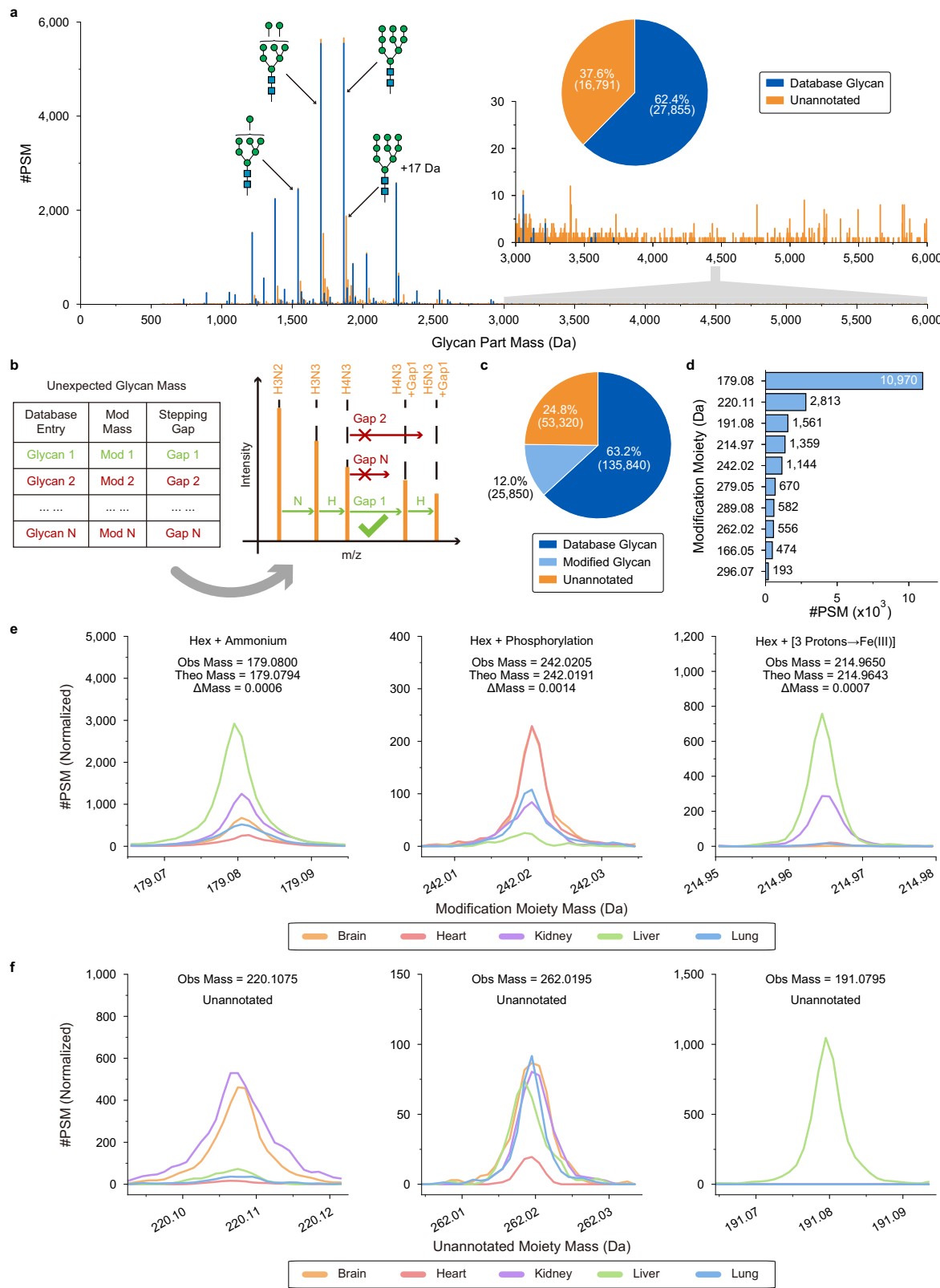

database entries, yielding a high proportion of PSMs (37.6%, 16,791) with unannotated glycans (Fig. 3a). To deduce the saccharide compositions of the unannotated glycans, we developed a monosaccharide stepping method (Fig. 3b and Supplementary Fig. 16) to match Y ions from the stepwise fragmentation of glycans. In addition to the basic saccharide composition of glycans (e.g., Hex(7)HexNAc(2)), the mass of unexpected glycan

modifications was also determined. Supplementary Fig. 17 presents two spectrum examples in which the compositions of Hex(5)HexNAc(2) with the moiety of 179 Da and Hex(6)HexNAc(2) with the moiety of 242 Da were deduced by monosaccharide stepping strategy. By using this method, we revealed the glycan compositions for 21.2% (9482/44,646) of the PSMs in the mouse liver dataset. From the five mouse tissue datasets, the

**Fig. 3 Elucidation of the modification moieties on glycans. a** Mass histogram of glycans in glycopeptides of the mouse liver dataset. The glycan masses were obtained by the glycan database-independent pipeline in Glyco-Decipher. Blue: PSMs with glycan mass that matched GlyTouCan glycans; Orange: PSMs with glycan mass that could not be annotated by GlyTouCan glycans. The bin width was set to 1 Da and centered at integer values. **b** A monosaccharide stepping method was designed for the deduction of the composition and the mass of modification moiety on modified glycans. **c** Percentage of PSMs with glycan parts annotated by database glycans and modified glycans. **d** Mass values of the ten most abundant modification moieties on glycans from glycopeptide spectra of the five mouse tissues. **e** Mass profiles of the modification moieties with known chemical compositions on modified glycans. **f** Examples of mass profiles of glycan modification moieties with unannotated composition. Source data are provided as a Source Data file.

glycans in 25,850 additional PSMs were uncovered, accounting for 12.0% (25,850/215,010) of the total PSM results (Fig. 3c).

Among the deduced modification moieties (Supplementary Fig. 18), some matched the monosaccharide (e.g., 163 Da for the isotopic-shifted mass of Hex), suggesting the addition of a corresponding monosaccharide to the deducted composition. Furthermore, there were 27 modification moieties (>20 GPSMs for each) that did not match any known monosaccharide (Supplementary Fig. 19), leading to the discovery of 164 modified glycans with mass values could not be annotated by GlyTouCan entries (Supplementary Data 3). The mass values for the 10 most abundant modification moieties are listed in Fig. 3d. The 179-Da moiety was the one with the most PSMs and was mostly added to oligo-mannose glycans (Supplementary Fig. 18). The observed mass of the 179 Da moiety coincided with the mass value of reported ammonium-adducted Hex (Hex+NH$_3$)[31,32] (Fig. 3e). The high similarity between the spectra of +179-Da-modified glycopeptides and their unmodified counterparts indicated that ammonium adduction occurs at the terminus of glycan structures (Supplementary Fig. 20), which is in line with a previous report[33]. Glyco-Decipher also demonstrated its ability to detect many other low-abundance glycan modifications. For example, glycans linked with a moiety of 242 Da (Hex + phosphorylation) (Supplementary Fig. 17), which is known as mannose-6-phosphate[34] (M6P), were detected in 1,144 glycopeptide spectra of the mouse tissues. As a key targeting signal in lysosomal hydrolase transfer[35,36], M6P exhibited distinct abundance distributions across the datasets of different tissues (Fig. 3e), implying disparate lysosomal activity in these tissues. Glycans with a moiety of 215 Da (Hex + 53 Da) were also detected (Fig. 3e), and intriguing dissociation of 53 Da from Y fragment ions was observed in MS2 (Supplementary Fig. 21), indicating that displacement of three protons by iron ions (3 + ) occurred on glycopeptides[23].

Glyco-Decipher also discovered many unassigned moieties on glycans, including +220 Da and +262 Da at the terminus of glycan structures[37,38] (Fig. 3f and Supplementary Fig. 22). After adopting the monosaccharide stepping method, ~24.8% of the PSMs remained with unannotated glycan composition due to the lack of sufficient Y fragment ions in their spectra (Fig. 3c and Supplementary Figs. 23 and 24). Some of them were deduced to have glycans of high mass values (Supplementary Fig. 24), suggesting high diversity of glycosylation. Overall, Glyco-Decipher can comprehensively profile the glycans attached to glycopeptides and enables the discovery of unexpected modified glycans (Supplementary Note 2 and Supplementary Figs. 49–52). The dedicated monosaccharide stepping method demonstrated its ability to reveal unexpected modifications on glycans which could not be achieved by other tools.

**Systematic evaluation of the performance of Glyco-Decipher in glycoproteomics analysis.** To evaluate the performance of Glyco-Decipher in large-scale N-glycoproteomics analysis, we systematically compared Glyco-Decipher with four software tools, including Byonic[8] (v3.11.3), MSFragger-Glyco[10] (v3.1.1), StrucGP[12] (v1.0.0), and pGlyco 3.0[13] (v20210615), by using the

dataset of mouse tissues (Supplementary Fig. 25). Notably, the peptide matching in Glyco-Decipher and StrucGP is glycan database-independent, while a list of glycans (pGlyco 3.0 and Byonic) or glycan masses (MSFragger-Glyco) are required for the other tools. Compared with the listed search tools, Glyco-Decipher reported the most PSMs and provided a 33.5–178.5% increase in the number of identified glycopeptide spectra due to the glycan database-independent search and spectrum expansion strategy (Fig. 4a, top). The consistency of peptide identification with the other tools was observed at both identification stages (i.e., before and after spectrum expansion) of Glyco-Decipher (Supplementary Fig. 26a). As a result, most of the glycopeptide spectra commonly identified by Glyco-Decipher and the other tools were matched to identical peptides. For example, 98.3% (69,349/70,576) of the spectra that commonly identified by Glyco-Decipher and StrucGP were matched to the same peptides. Glyco-Decipher also reported the most peptide identifications from the glycopeptide spectra of mouse tissues and covered 72% or more of the peptide sequences in paired comparisons with the other tools (Fig. 4a, bottom).

In terms of glycopeptide identification, we first investigated the confidence of these tools in multiple aspects (Fig. 4b). The dataset containing HCD spectra of glycopeptides from $^{13}$C/$^{15}$N metabolically labeled yeast[9] was searched using these tools with the same parameters ("Methods"). True-positive identifications would result in $^{13}$C/$^{15}$N peak pairs in MS1, which correlate to the number of carbon/nitrogen atoms in the elemental composition of glycopeptides[9]. From the matching results of isotopic peak pairs, only Glyco-Decipher, StrucGP and pGlyco 3.0 correctly estimated the FDR of glycopeptide identification at the spectrum level (line plot in Fig. 4b and Supplementary Data 4). In the analysis of glycan compositions, glycans in almost all the GPSMs (>98.5%, pie plot in Fig. 4b) reported by the three tools agreed with known biosynthesis rules[39], in which only oligo-mannose glycans with the composition of Hex(n)HexNAc(2) are linked on yeast proteins. The above analysis indicated that reliable glycan assignments were achieved in Glyco-Decipher and StrucGP/pGlyco 3.0 by comprehensively matching B/Y ions of glycans. Additional FDR analysis of the results gained from spectrum expansion also suggested the high confidence of Glyco-Decipher in glycopeptide identification with the adoption of peptide fragmentation pattern (Supplementary Fig. 26b). However, the number of interpreted glycopeptide spectrum of StrucGP is significantly less than that of Glyco-Decipher and pGlyco 3.0 on the yeast dataset (Fig. 4b), because the design of StrucGP is mainly for human and mammalian models[12]. Higher proportions of incorrect glycans were reported by Byonic (variable modification search) and MSFragger-Glyco (mass offset search) from the proteome-scale glycan database due to only mass values of glycans were utilized in matching (pie plot in Fig. 4b and Supplementary Fig. 27).

In addition to database glycan matching, Glyco-Decipher retained high confidence in modified glycan discovery. The identification of modified glycans could also be achieved in pGlyco 3.0 by enumerating all possible modified forms of

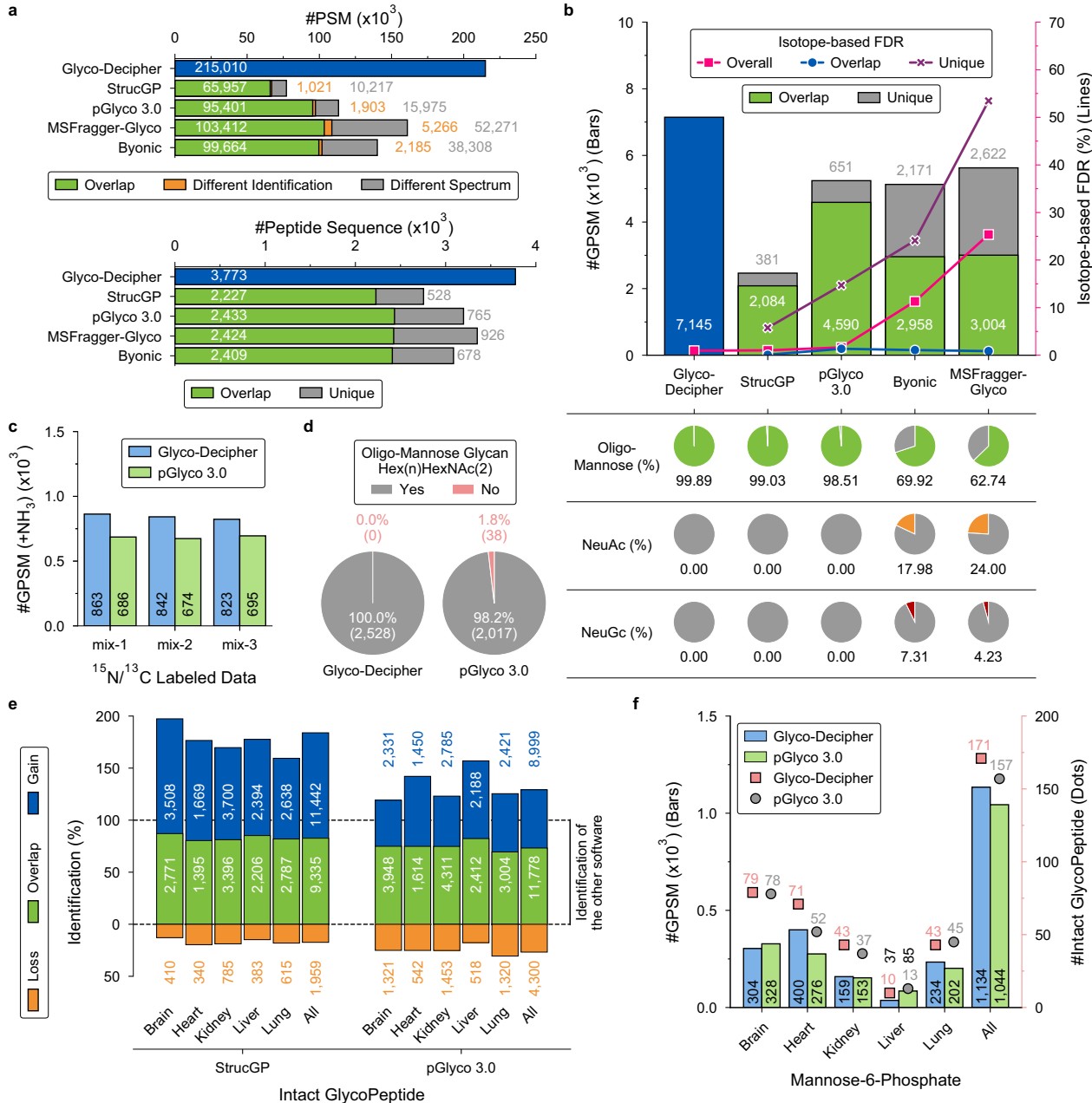

**Fig. 4 Comparison between Glyco-Decipher and other software tools. a** Comparison of the performance between Glyco-Decipher, StrucGP, pGlyco 3.0, MSFragger-Glyco and Byonic using the dataset of mouse tissues. Top: Comparison between Glyco-Decipher and other software tools in the performance of glycopeptide-spectrum interpretation. Green: the spectra matched to identical peptide backbones in Glyco-Decipher and another tool. Orange: the spectra commonly identified in Glyco-Decipher and another tool but matched to different peptide backbones. Gray: spectra specifically matched by another tool. Bottom: Comparison between Glyco-Decipher and other software tools in peptide sequence identification. Green: the peptide sequence commonly identified by Glyco-Decipher and another tool in pair comparison. Gray: the peptide sequence specifically matched by another tool in pair comparison. **b** FDR analysis using the $^{13}C/^{15}N$ metabolically labeled yeast dataset. The isotope-based FDR was calculated by matching $^{13}C/^{15}N$ isotopic peak pairs in MS1 spectra (line). Red line: FDR analysis based on the total GPSM results of each tool; blue line: FDR analysis based on the GPSM results that overlapped with Glyco-Decipher; purple line: FDR analysis based on the GPSMs specifically identified by each tool. The proportions of GPSMs of oligo-mannose glycans with composition of Hex(n)HexNAc(2) (green pie), NeuAc (orange pie) or NeuGc (red pie) containing glycans identified by each tool are shown in the bottom table. Source data are provided as a Source Data file. **c** Distributions of GPSMs of ammonium-adducted glycans reported by Glyco-Decipher and pGlyco 3.0. **d** Number of ammonium adduction GPSMs with/without oligo-mannose glycan composition. **e** Comparison of glycopeptide identification results between Glyco-Decipher and StrucGP/pGlyco 3.0. The additional (gain), overlap and lost identifications of Glyco-Decipher compared to other software tools are indicated by blue, green and orange bars, respectively. **f** Distributions of mannose-6-phosphate (M6P) GPSMs (bars) and intact glycopeptides (dots) across mouse tissues reported by Glyco-Decipher and pGlyco 3.0. All M6P identifications were validated by the diagnostic oxonium ion (phosphorylated hexose, $m/z = 243.0269$) in the glycopeptide spectra.

database entries with user-provided monosaccharide modification. From the yeast dataset, 2528 and 2055 GPSMs of ammonium-adducted glycans were identified in Glyco-Decipher and pGlyco 3.0, respectively (Fig. 4c). Note that all the ammonium-adducted glycans matched in Glyco-Decipher are oligo-mannose glycans, which is in line with the glycosylation rule in yeast. In contrast, 1.8% of the ammonium adduction spectra reported by pGlyco 3.0 were matched to non-oligo-mannose glycans, e.g., Hex(11)HexNAc(2)Fuc(1) (Fig. 4d). The analysis of the identification results indicated that the incorrect glycan assignments in pGlyco 3.0 may originate from the largely increased glycan search space of the enumeration method (Supplementary Fig. 28). Benefiting from glycan database-independent searching, Glyco-Decipher enables the identification of modified glycans in discovery mode and the stepwise Y ion matching in monosaccharide stepping also ensures reliable elucidation of modified glycan compositions.

The performance of Glyco-Decipher was further compared with StrucGP and pGlyco 3.0 in detail due to their comparable confidence in glycopeptide identification. Taking advantages of the sensitivity of spectrum expansion, Glyco-Decipher reported more glycopeptide identifications even when only the results with GlyTouCan glycans were considered (Supplementary Fig. 29a), suggesting improved coverage of glycosylation analysis based on database glycan compositions. Combined with the results of discovered modified glycan, Glyco-Decipher yielded a total of 20,777 unique intact glycopeptide identifications from mouse tissues, leading to increases of 84.0% and 29.2% compared to StrucGP and pGlyco 3.0, respectively (Fig. 4e). Similar distributions of glycans in different categories were observed in the results of Glyco-Decipher and pGlyco 3.0 (Supplementary Fig. 29b). However, compared to the results of Glyco-Decipher and pGlyco 3.0, only ~1/3 complex/hybrid glycans were identified by the modularization strategy in StrucGP due to the increased complexity of glycan ions in glycopeptide spectra, leading to a biased interpretation of glycans (Supplementary Fig. 29b). Nevertheless, the modularization strategy is of great potential to interpret glycopeptide spectra with unexpected glycans beyond database entries. Apart from searching with GlyTouCan glycans, phosphorylation on hexose was set in pGlyco 3.0, and monosaccharide stepping was performed in Glyco-Decipher to evaluate their performance in identifying low-abundance modified glycans. Benefit from glycan-independent peptide matching, Glyco-Decipher is able to profile glycans in complex samples (Supplementary Fig. 13) and most of the unexpected/modified glycans identified by StrucGP/pGlyco 3.0 from the mouse dataset were also successfully detected by Glyco-Decipher (Supplementary Fig. 29c). The detailed comparison of the glycopeptide results of M6P glycans also demonstrated the high sensitivity of Glyco-Decipher in glycoproteomics analysis: pGlyco 3.0 reported 157 M6P glycopeptides in 1044 spectra by enumerating phosphorylation on database glycans (Fig. 4f). In contrast, phosphorylation on hexose was confidently detected in 1134 glycopeptide spectra with diagnostic oxonium ion (phosphorylated hexose, $m/z = 243.0269$) by Glyco-Decipher, leading to the identification of 171 M6P glycopeptides (Fig. 4f and Supplementary Fig. 29d).

To further evaluate the ability of Glyco-Decipher in unveiling the heterogeneity of glycosylation in complex samples, glycopeptides were enriched from the tryptic digest of a human serum sample and analyzed by LC–MS/MS under 80, 160, and 210 min reversed-phase chromatographic gradients, respectively. The tandem mass spectra were collected upon stepped-energy HCD fragmentation at 20–30–40% with an Orbitrap Exploris 480 ("Methods"). We then evaluated the performance of Glyco-Decipher with the acquired serum data (Supplementary Fig. 30a, b). In total, Glyco-Decipher provided 132.4% and 51.8% more

interpreted glycopeptide spectra compared to StrucGP and pGlyco 3.0, respectively. The improvements in spectrum interpretation also elevated the glycoproteome coverage of human serum, leading to the identification of 6478 site-specific glycans in human serum, including 1523 modified site-specific glycans (Supplementary Data 5). Glycoproteins in human serum are heavily sialylated, which is a useful feature in clinical application for the detection, staging and prognosis of different diseases and cancers[40]. Compared to StrucGP and pGlyco 3.0, more sialic acid-containing site-specific glycans, including the multi-sialylated ones, were uncovered by Glyco-Decipher on the serum glycoproteins (Supplementary Fig. 30c, d). In addition, NeuGc-containing glycans are negligible for the glycosylation of human serum[41] (Supplementary Fig. 31a), and the feature has also been used to evaluate the accuracy of glycopeptide identification[14]. The absence of NeuGc-containing glycan in the results of Glyco-Decipher also suggests its reliability in glycan assignment (Supplementary Fig. 31b).

To assess the capability of Glyco-Decipher in benefiting biological research by offering deeper coverage of glycosylation heterogeneity, we also reanalyzed N-glycosylation datasets of the SARS-CoV-2 spike and its receptor angiotensin-converting enzyme 2 (ACE2) from a recent study by Zhao et al.[42]. The SARS-CoV-2 coronavirus is responsible for severe acute respiratory syndrome and utilizes a spike glycoprotein trimer for host cell receptor binding[43]. ACE2, known as a receptor of SARS-CoV-2 in the human body[44], is also a glycoprotein that acts as a regulator of cardiovascular homeostasis. Reanalysis of the MS data revealed that Glyco-Decipher identified 1640 and 310 site-specific glycans on the SARS-CoV-2 spike and ACE2, respectively, which is an ~70% increase in the identification number compared to that of the original study in which the data were searched with pGlyco 2.0[9] (Supplementary Fig. 32a, Supplementary Data 6). More glycan identifications were reported by Glyco-Decipher on all 22 glycosites of the SARS-CoV-2 spike and 6 glycosites of ACE2 (Supplementary Fig. 32b, c). It should be noted that glycans containing NeuGc were not observed in the identification results of Glyco-Decipher (Supplementary Fig. 33a), which is in line with the glycosylation rule of SARS-CoV-2 spike protein cultured in HEK293 T cell[42]. More detailed information on the site-specific glycans identified on SARS-CoV-2 spike and ACE2 are provided in Supplementary Fig. 33. Extended comparison with StrucGP/pGlyco 3.0 on this dataset also demonstrates the improvements of Glyco-Decipher in glycoproteomics analysis (Supplementary Fig. 34).

**Glyco-Decipher allows determination of the abundance distribution of site-specific glycans.** In addition to the identification of intact glycopeptides, quantitation of glycopeptides is another essential issue in assessing glycosylation levels for functional studies. A quantification module was developed and embedded in Glyco-Decipher. In this module, the theoretical isotopic peak list of the precursor was calculated based on the elemental composition of the intact glycopeptide. The elution profile of the precursor was extracted from the MS1 spectra around its retention time, and the area under the elution profile was used for quantification (Supplementary Fig. 35). This module enables the determination of the abundance distribution of each glycan or a specific glycosylation type in a system (Supplementary Fig. 36). Correlation analysis of glycosylation at the sample level or the site level revealed the similarity of glycosylation between systems (Supplementary Figs. 37 and 38).

The occupation of diverse glycans on a glycosite is under the regulation of glycosyltransferases and glycosidases and the abundance distribution of site-specific glycans can be readily

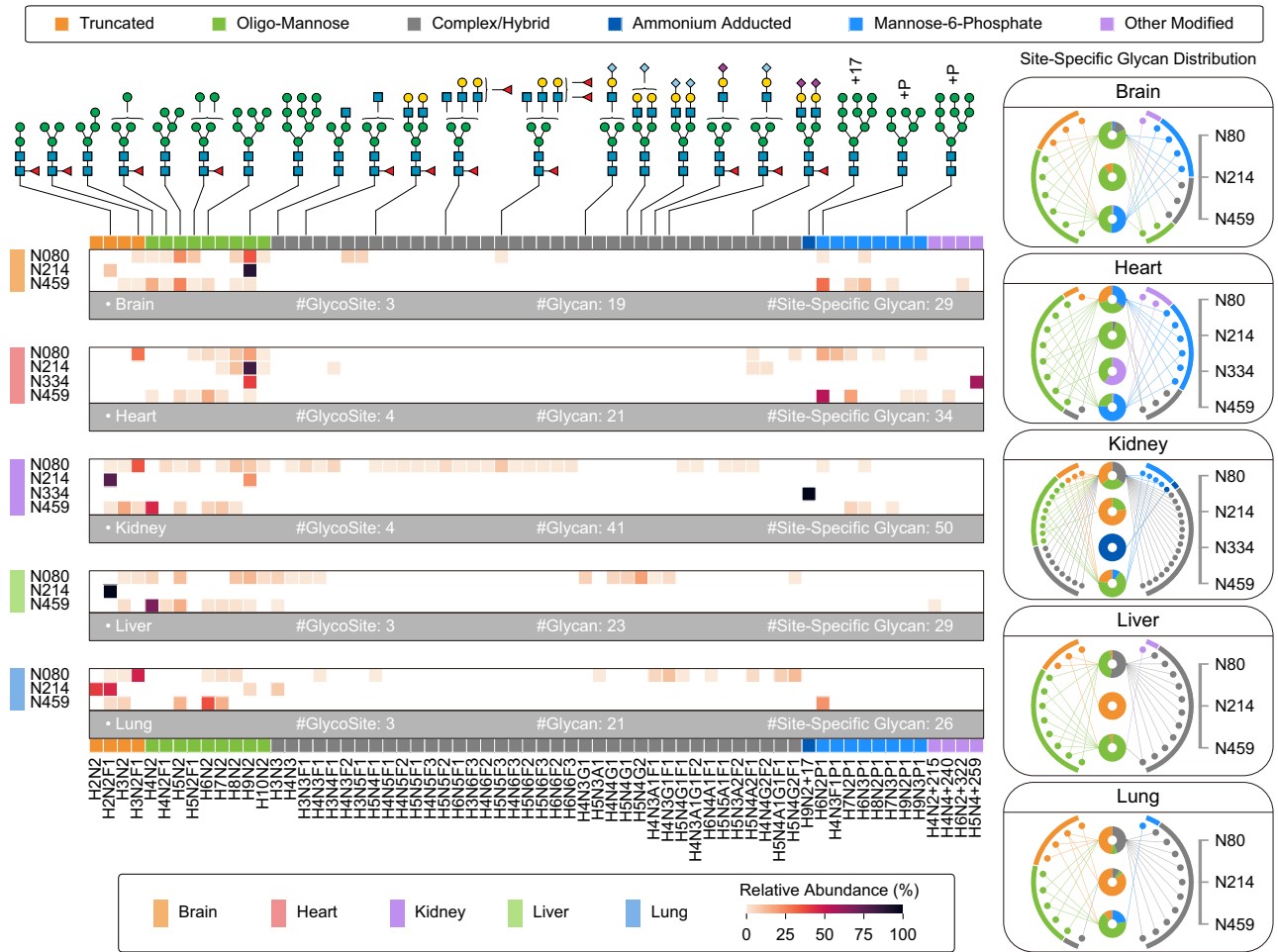

**Fig. 5 Quantitative analysis of site-specific glycosylation on prosaposin by Glyco-Decipher.** Relative abundance distribution of glycans at each glycosite in prosaposin across five mouse tissues (heat map). Compositions and possible structure illustration of the high-abundance glycans are annotated at the top. For a more intuitive demonstration, the abundance distributions of glycans at each glycosite are listed in the right radial diagrams. In each radial diagram, nodes around the circle denote glycans linked to prosaposin, and donuts in the center denote glycosites identified in prosaposin. Linkage between the node and the center donut indicates that the glycosite was modified by the corresponding glycan. The percentage value in each donut indicates the relative abundance for a certain type of glycan. All M6P identifications were validated by the diagnostic oxonium ion (phosphorylated hexose, $m/z = 243.0269$) in glycopeptide spectra. H Hex, N HexNAc, A NeuAc, G NeuGc, F Fuc, P Phosphorylation. See Supplementary Data 7 for detailed information of glycan nodes.

determined by Glyco-Decipher. Prosaposin (UniProt: Q61207), the precursor of saposin A-D, which acts as a cofactor for lysosomal hydrolysis of sphingolipids, was exemplified in detail (Supplementary Data 7). As presented in Fig. 5, the abundance distributions of different types of glycans at glycosites in prosaposin are illustrated. Complex/hybrid glycans of ten distinct compositions, account for a large proportion (43.9%) of glycosylation at glycosite N80 of prosaposin in the lung, including those with relatively low abundance (e.g., Hex(4)HexNAc(3)NeuAc(1)Fuc(1) of 5.4%). In contrast, less than five complex/hybrid glycans were identified by StrucGP and pGlyco 3.0 at glycosite N80 in the lung, thus leading to a biased distribution calculation (Supplementary Figs. 39 and 40).

Notably, modified glycans that are missed in glycan database-dependent tools also contribute to the determination of distributions of site-specific glycans. For example, the ammonium-adducted glycan was specifically identified by Glyco-Decipher at N334 of prosaposin in the kidney (Fig. 5 and Supplementary Fig. 41). M6P glycans, which are associated with lysosomal trafficking of prosaposin via the cation-independent mannose-6-phosphate receptor[45] (M6PR), were

detected with relatively high abundance at the glycosite N459 in brain and heart. For example, M6P glycans accounted for 74.1% of the abundance distribution at this site in mouse heart, as determined by Glyco-Decipher (Fig. 5). Similar abundance distributions of M6P glycans were observed based on the results of pGlyco 3.0 (Supplementary Fig. 40). However, despite their relatively high abundance, these modified glycans were not identified by the other glycan database-dependent tools. Overall, the in-depth coverage of glycosylation heterogeneity of Glyco-Decipher offers more accurate relative quantification of site-specific glycosylation.

## Discussion

Current glycoproteomics software tools typically determine peptide backbones based on a single mass spectrum. Only a few studies attempted to utilize the connections between the spectra of glycopeptides with the same peptide backbones to improve glycopeptide identification. For example, the MS1 features of glycopeptides with identical peptide backbones, e.g., differences in precursor mass and retention time, were exploited to enhance the

identification performance[46]. However, this strategy is prone to yield more false-positive identifications as the MS2 fragmentation information is not used[47]. Spectral clustering based on global MS2 spectrum similarity was also reported to assist glycopeptide identifications[48], yet the diverse glycan modifications and the predominance of glycan fragment ions in N-glycopeptide spectra limit its sensitivity. In Glyco-Decipher, the in silico deglycoylation strategy is adopted to remove the interference of glycan ions firstly, and then the high similarity of peptide fragmentation patterns among glycopeptides sharing the same peptide backbone is used to match spectra with poor peptide fragmentation, which significantly improves the sensitivity of glycopeptide identification. In the glycoproteomics dataset of mouse tissues, a 44.5% increase in the number of site-specific glycans of complex/hybrid type was achieved after spectrum expansion. This is probably because the abundance of this type of glycans is relatively lower than that of oligo-mannose glycans in the dataset of mouse tissues. In addition, negatively charged sialyl residues in sialylated glycans hamper the fragmentation of peptide backbones and lead to insufficient peptide fragments in glycopeptide spectra. The consistent peptide fragmentation pattern of the shared peptide utilized in Glyco-Decipher overcomes this barrier and provides deeper insight into glycosylation heterogeneity.

Most existing glycoproteome software tools (e.g., Byonic, MSFragger-Glyco, and pGlyco 3.0) rely on glycan databases to decipher glycopeptide spectra and are unable to interpret the spectra of glycopeptides with glycans not present in databases. As an alternative approach, open search is able to identify peptide backbones without setting any modifications and makes the discovery of unexpected glycans possible. However, the wide mass distribution of N-glycans (from hundreds to over 6000 Da) substantially expands the search space of open search in candidate peptide matching, leading to limited identification sensitivity and potential random matches. Benefiting from the fragmentation features of the N-glycan core structure and peptide backbones, Glyco-Decipher offers more sensitive and accurate glycan database-independent peptide matching than the open search method. The recently introduced StrucGP adopts a modularization strategy for the structural interpretation of site-specific N-glycans by matching B/Y ions against a built-in database of glycan core/branch structures. It allows the discovery of glycans with unexpected structures but is unable to cover glycans with additional modification moiety. Also, the complexity of glycan fragmentation and the lack of sufficient B/Y ions in glycopeptide spectra limit its sensitivity in the identification of complex/hybrid glycans. In addition to matching with database glycans, glycans with modifications could be searched in pGlyco 3.0 by enumerating all modified forms of database entries with user-provided monosaccharide modifications. In contrast, the database-independent searching and monosaccharide stepping in Glyco-Decipher permit the discovery of unexpected monosaccharide modifications and provide deeper insight into the glycosylation heterogeneity of modified glycans. Detailed analysis of the identification results on the yeast dataset (Fig. 4d and Supplementary Fig. 28) also indicated that the monosaccharide stepping method ensures higher accuracy in modified glycan identification than the enumeration method.

Indeed, additional work is still needed to improve Glyco-Decipher. First, Glyco-Decipher is currently designed for the interpretation of N-glycopeptide spectra acquired with sceHCD fragmentation and supports to accommodate other fragmentation methods and O-glycosylation analysis will be provided in the near future. Second, improvements for the detection of precursors, especially for chimeric ones, are needed for the glycoproteomics tools including Glyco-Decipher (Supplementary Figs. 42 and 43). Third, the determination of fine glycan structures is unable to be

achieved in Glyco-Decipher and this could be benefited from glycomics methods, which have been increasingly used in glycosylation analysis[49–51]. Moreover, rare but important issues in glycopeptide identification, including the identification of N-glycans lacking sufficient core structure ions (e.g., GlcNAc(1)) and the ambiguity in localization of labile modifications on peptides and glycans, are needed to be addressed.

In summary, we introduced a glycoproteomics analysis tool, Glyco-Decipher, which enables database-independent profiling of N-glycans and provides a more comprehensive characterization of protein glycosylation than existing software tools. Powered by spectrum expansion and glycan database-independent searching, at least 34% more glycopeptide spectra were interpreted by Glyco-Decipher compared to other tools. Beside entries in the GlyTouCan, the glycan database-independent search scheme unveiled 164 modified glycans with 27 diverse modification moieties in mouse tissues. The glycan modification moieties revealed by Glyco-Decipher provide a rich resource for further glycobiology studies. With a friendly graphical user interface (GUI) (Supplementary Fig. 44), the freely available Glyco-Decipher has the potential to become a valuable tool for proteome-wide studies of glycosylation.

## Methods

### Collection of the human serum dataset

*Materials and samples.* Acetonitrile (ACN) was provided by Merck (Darmstadt, Germany). Ammonium bicarbonate ($NH_4HCO_3$), urea, dithiothreitol (DTT), iodoacetamide (IAA), trifluoroacetic acid (TFA), and trypsin (bovine, TPCK-treated) were obtained from Sigma (St. Louis, MO). Formic acid (FA) was purchased from Fluka (Buches, Germany). Other chemicals and reagents were of or above analytical grade. High-purity water used in all experiments was obtained from a Milli-Q purification system (Millipore, Milford, MA).

*Protein digestion of human serum.* The human serum sample collection was approved by the Ethical Committee of Xijing Hospital, Fourth Military Medical University, Xi'an, China (Approved No. of the ethic committee: KY20192088-F-1). Written informed consent was obtained from all participants. The study design and conduct complied with all relevant regulations regarding the use of human study participants and was conducted in accordance to the criteria set by the Declaration of Helsinki. The human serum samples were collected from 48 gastric cancer patients, pooled into a single sample, and stored at −80 °C. The proteins in the pooled serum sample were denatured in 8 M urea/100 mM $NH_4HCO_3$, reduced by dithiothreitol followed by alkylated with iodoacetamide. Next, tryptic digestion was carried out at 37 °C for 18 h and terminated by adding trifluoroacetic acid. The peptides were desalted with C18 solid-phase extraction (SPE) cartridges (Waters, MA), dried down, and stored at −20 °C for further use.

*Enrichment of intact glycopeptides.* Isolation of intact glycopeptides was conducted on a Waters Acuity UPLC system (Waters, USA) equipped with a homemade HILIC (hydrophilic-interaction chromatography) column (click maltose[52], 5 µm, 100 Å) according to our recently established automated method[53]. Briefly, peptides digested from 5 µL serum were dissolved in 35 µL 80% ACN/1% TFA and loaded onto HILIC column. Mobile phase A and B were 0.1% TFA in $H_2O$ and 98% ACN/0.1% TFA, respectively. The gradient was kept at 80% mobile phase B for 11 min and then dropped directly to 30% mobile phase B and maintained for 5 min. Intact glycopeptides were collected during 12–13 min of the gradient. The sample was dried for LC–MS/MS analysis.

*Liquid chromatography-tandem mass spectrometry analysis.* LC–MS/MS analysis was performed on an EASY-nLC™ 1200 system (Thermo Fisher Scientific, USA) coupled with an Orbitrap Exploris 480 mass spectrometer (Thermo Fisher Scientific, USA). The analytical column (75 µm i.d.) was packed with C18-AQ beads (1.9 µm, 120 Å) to 35 cm length. Mobile phase A and B were 0.1% FA in $H_2O$ and 0.1% FA in ACN respectively. Peptides were separated under three different gradients, i.e., 80, 160, and 210 min. For the 80-min gradient, mobile phase B was increased linearly from 9 to 45% in 68 min, followed by 45 to 90% for 2 min and maintained at 90% for another 10 min. For the 160-min and 210-min gradients, linear increase of mobile phase B from 9% to 45% in 138 min and 188 min were executed, respectively. Intact glycopeptide were dissolved in 0.1% FA and loaded onto the analytical column for LC–MS/MS analysis. The mass spectrometer was operated in data-dependent mode. Full mass scan MS spectra (*m/z* 400–2000) were acquired by the Orbitrap mass analyzer with a 60,000 resolution. Intact glycopeptides were analyzed with a 300% normalized AGC target with a maximum injection time of 40 ms. RF lens was set as 45%. MS/MS scans were also acquired by

the Orbitrap mass analyzer with a 30,000 resolution, and the normalized AGC target was set to 500% with a maximum injection time of 60 ms. Isolation window was set as 1.4 $m/z$. HCD fragmentation was performed with 20–30–40% stepped NCE. Tune Application (Thermo Fisher Scientific, v3.1.279.9) and Xcalibur software (Thermo Fisher Scientific, v4.4.16.14) were used for the control of mass spectrometer and data collection.

**Data preparation.** In addition to the dataset of human serum, several publicly available N-glycoproteomics datasets were downloaded from the PRIDE[54] data repository, with the accession numbers PXD005411, PXD005413, PXD005412, PXD005553, PXD005555 (Liu et al.[9] mouse tissues, including the brain, heart, kidney, liver, and lung), PXD005565 (Liu et al.[9] [13]C/[15]N metabolically labeled yeast) and PXD019937 (Zhao et al.[42] SARS-CoV-2 spike and ACE2). All raw files were converted to the open-source format mzML[55] by using ProteoWizard[56] (version 3.0.21105) with 32-bit precision and the "1-" peak picking option.

**Glyco-Decipher algorithm**

*Peptide backbone mass derivation and in silico deglycosylation of intact glycopeptide spectra.* Only spectra containing more than two oxonium ions (Supplementary Table 2) were processed as the spectra of intact glycopeptides. After the removal of oxonium ions, the $m/z$ gaps of the pentasaccharide core structure fragment ions (i.e., Y0, Y-HexNAc(1), Y-HexNAc(2), Y-Hex(1)HexNAc(2), Y-Hex(2)Hex-NAc(2), Y-Hex(3)HexNAc(2)) were matched at different charge states (from +1 to the precursor charge). At least three core structure Y ions were used to screen the matched core structure candidates. To avoid incorrect assignment of the glycosylation type of small glycans (e.g., GlcNAc(1)) in HCD spectra, small glycans without Y2 ion were not considered in this work and could be covered by decreasing the number of required core structure ions when searching with Glyco-Decipher. After merging isotopic forms, core structures were scored and sorted by the intensity of Y1 ions if more than one possible peptide mass value was deduced from the matched core structures in one spectrum. After the derivation of the peptide mass, ions of the corresponding core structure were removed. To reduce the interference from other Y ions for peptide backbone identification, the fragment ions at the high $m/z$ section were removed if the $m/z$ difference between it and the previously matched Y ions equaled to any monosaccharide mass; this stepping process was executed until no more fragment ions matched. After all the oxonium ions and Y ions were removed, the in silico deglycosylated spectrum was finally generated by overwriting the precursor $m/z$ with the $m/z$ corresponds to the mass of (peptide + HexNAc) for the consideration of (b/y + HexNAc) ions.

*Spectrum expansion after in silico deglycosylation.* The newly generated spectrum file was searched against a motif (N-X-S/T/C, where X is not P)-containing peptide database with the search engine MS-GF+ [27], which is an open-source software and is already integrated in Glyco-Decipher. Spectrum expansion was applied to increase the spectrum identification rate of intact glycopeptides. From the MS-GF + identification results, PSMs with a $q$ value <0.01 were retained as high confidence results (the $q$ value was defined as the minimum FDR at which the identification was called significant in MS-GF+ ). The experimental spectrum was matched to the peptide's theoretical fragmentation ions, intensity values of the same types of ions at different charge states (+1/+2) were summed, and the peptide fragmentation pattern in a PSM was generated by normalizing the intensity of each type of ion against the maximum intensity value. A peptide's average fragmentation pattern was calculated by averaging all the corresponding PSM patterns in the glycopeptide spectra with the same peptide backbone. Then, unidentified spectra around each peptide's average retention time were matched to the peptide (the retention time window was set to be 1800 s in this work). The obtained average peptide fragmentation pattern was used in spectrum expansion scoring:

$$\text{Score}_{\text{PSM}} = \text{Score}_{\text{Peptide}} + \text{Score}_{\text{Core}} = \text{coefficient} \times \sum_{i=1}^{n} \left(\frac{\text{Intensity}_i}{\text{frequency}_i}\right)^{\alpha} + \sum_{j=1}^{m} \left(\frac{\text{Intensity}_j}{\text{frequency}_j}\right)^{\alpha} \quad (1)$$

Where intensity is the relative intensity of the matched peptide fragment ions; frequency is the number of fragment ions generated by other peptides in the mass tolerance window; and the coefficient is the cosine similarity between the fragmentation pattern in the PSM and the average peptide pattern (this value should be near 1 if two patterns match well or 0 if a total random match occurs). $\alpha$ was set to 0.3 in this article; $n$ is the number of matched fragment ions of the peptide backbone, and m is the number of matched fragment ions of the N-glycan core structure (i.e., Y0, Y-HexNAc(1), Y-HexNAc(2), Y-Hex(1)HexNAc(2), Y-Hex(2)HexNAc(2), Y-Hex(3)HexNAc(2)). The score to assess a peptide-spectrum match consists of two parts: the score of peptide ions, which is to assess the peptide identification with the matched peptide ions and their intensity pattern; the score of core structure ions, which aims to avoid identification of peptides with incorrect mass, especially to avoid the incorrect assignment of peptides with shared peptide fragment ions (e.g., peptides with a different number of missed cleavages).

For quality control in spectrum expansion, decoy spectra ($m/z$ value of each fragment ion in MS2 was shifted with 1–30 $m/z$ randomly) and linear tail-fit method[28] were used to calculate the e-value, i.e., the expectation value that the obtained match is valid, of each PSM in spectrum expansion. Then all PSMs,

including target and decoy PSMs, are sorted in decreasing order of spectral e-values. And the FDR for PSMs with e-value ≤ a threshold $t$ is calculated with the formula of FDR = $(2 * N_{decoy})/(N_{decoy} + N_{target})$, where N is the number of PSMs with e-value ≤ $t$. Instead of setting a fixed score threshold for all peptides, the score threshold is calculated dynamically based on the e-value method during the spectrum expansion process of each peptide. Then PSMs with FDR < 0.01 and with at least three matched core structure fragment ions (in which the Y1 (peptide +HexNAc) ion is required) were retained for further glycan identification.

*Glycan annotation by database matching and monosaccharide stepping.* After peptide backbones were confidently identified, the precursor $m/z$ value was corrected to the mono-isotopic value by calculating the similarities between elution profiles of isotopic peaks, as well as the similarity between experimental intensity patterns and theoretical pattern. The calculation of similarities between elution profiles of isotopic peaks enables the removal of noise peaks. And the calculation of similarities between experimental peak pattern and theoretical pattern enables the determination of $m/z$ value of mono-isotopic peak. The theoretical isotopic pattern of glycopeptide was calculated with the knowledge of the composition of identified peptide backbone and the simulated composition of the glycan part. The pseudo-glycan composition contains known element composition of N-glycan core structure (HexNAc(3)Hex(2), 892 Da) and simulated composition for additional mass (glycan mass-892 Da) with Hex ($C_6H_{10}O_5$). This procedure is also applicable for glycans with mass less than 892 Da, for which the composition is simulated with negative additional mass. The precursor correction step aims to accurately determine glycan mass. After subtracting peptide mass from corrected precursor mass, precise glycan mass was obtained without the need for glycan databases. To assign the detailed composition of each mass value, a glycan database containing 10,936 entries corresponding to 1766 unique glycan compositions was extracted from GlyTouCan (https://glytoucan.org/). After glycan part mass calculation, theoretical fragment ions, including B and Y ions, of each candidate glycan in the mass tolerance window were matched to the glycopeptide spectra. Theoretical B/Y fragment ions were deduced from the fragmentation of glycan structures encoded in WURCS 2.0[57] format in the database. Only the top-scored glycan and the corresponding structure information were outputted as the glycan part identification result. The function below was used to score the glycan candidates based on matched glycan fragment ions between theoretical value and the experimental value:

$$\text{Score}_{\text{Glycan}} = \sum_{i=1}^{n} \left(\frac{\text{Intensity}_i}{\text{frequency}_i}\right)^{\alpha} \quad (2)$$

where intensity is the relative intensity of the matched fragment ions and frequency is the number of glycan ions generated by other glycopeptides in the mass tolerance window.

The monosaccharide stepping method was applied to reveal the monosaccharide composition of modified glycans that did not match any database entries. In this method, the mass of each database glycan was subtracted by the observed glycan mass to obtain the mass of possible modification on glycans. For example, a glycan with a mass of 1395.50 Da could be interpreted by a modification with a mass of 17.02 Da on H6N2 (1378.48 Da) or 33.02 Da on H5N2F1 (1362.48 Da). After the enumeration of candidate modification mass values on glycans, a stepping gap list was obtained from monosaccharide types to step glycan fragment ions from Y-Hex(3)HexNAc(2) to the terminal of intact glycopeptide (intact glycopeptide ions at different charge states were considered the stepping terminals). Only the stepping route that reached the terminal with the given stepping gaps was retained to deduce the monosaccharide composition and corresponding modified moiety mass on glycans. An incorrect moiety mass was removed if the corresponding stepping gap did not match any two continuous Y ions. The glycan composition from the monosaccharide stepping result with the maximum route number was retained as a highly confident result. The composition of the glycan in different PSMs was set to be identical if the corresponding glycan mass values were in the mass tolerance window. The information of identified glycan composition and linked modification moiety were output in a new result file.

For FDR evaluation in glycan annotation, a decoy spectrum was generated for each GPSM ($m/z$ value for each fragment ions in MS2 was shifted with 1–30 $m/z$ randomly). Theoretical fragment ions of the matched glycan were matched to the shifted experimental fragment ions for the assessment of the possibility that the glycan-spectrum matching is random. The expectation value was calculated for each glycan-spectrum match via linear tail-fit method[28] and the FDR of the glycan identification results was calculated.

For the five mouse tissue datasets, after peptide identification and precursor correction, the calculated glycan mass values were collected to generate a glycan mass histogram. For rough mass distribution, each bin was set to be 1 Da in width ranging from 0 Da to 6000 Da. For the derivation of precise mass bins of glycans and modification moieties, sliding windows of 0.002 Da and 0.001 Da adopted in PTM-Shepherd[58] were used for peak picking. For detailed mass profiling of obtained moieties on glycans, the bin width was set to 0.001 Da, and the smoothed method used in Kong et al.[23] was adopted to obtain a monotonic peak shape. The center of the bin with the maximum PSM number was set to be the observation mass value.

*Glycopeptide quantification.* After intact glycopeptides were identified, elution profiles of each glycopeptide at different charge states were extracted from the mzML file with the calculated isotopic peak list in MS1. Mass tolerance in MS1 was set to 20 ppm. Glycopeptides with modified glycans were also considered in quantification, and their theoretical peak lists were deduced from the calculated molecular mass. Due to glycopeptide enrichment, the calculation of site occupancy is not applicable in this work. With the extracted elution profile of each glycopeptide, the glycan relative abundance was determined by normalizing the elution area at site-specific glycoform levels as well as the total glycosylation of the proteins.

**Performance evaluation of in silico deglycosylation with mouse dataset**. All searches were performed on a desktop computer running Windows 10 Pro (v.20H2), with two 2.20 GHz Intel Xeon E5-2650 v4 CPU processors with 64 Gb of installed RAM.

Unique peptides refer to unique amino acid sequences and modification linked to them. Unique (intact) glycopeptides refer to unique peptide backbones and attached glycans linked on them; that is, glycopeptides with identical peptide backbone but different glycans refer to different intact glycopeptide identifications. Unique site-specific glycans refer to unique combinations of glycosites and glycans linked on them; that is, site-specific glycans with identical glycosite but different glycans refer to different site-specific glycans. In this work, the glycans were classified into three categories based on their compositions: truncated glycans (Hex(<4)HexNAc(<3)Fuc(<2)), oligo-mannose glycans (Hex(>3)HexNAc(2) Fuc(<2)) and complex/hybrid glycans. In addition, complex/hybrid glycans were divided into four subgroups based on the glycan compositions: fucose and sialic acid-free glycans (no fucose and sialic acid in glycan composition); fucosylated but sialic acid-free glycans; sialylated but fucose free glycans and glycans containing both fucose and sialic acid.

LC–MS/MS datasets of five mouse tissues[9] (brain, heart, kidney, liver, and lung) were used to investigate the peptide mass determination and in silico deglycosylation performance of Glyco-Decipher. In comparison with MAGIC, MS2 mass tolerance was set to 0.02 Da because ppm was not supported in MAGIC. MS2 mass tolerance of 20 ppm was set in Glyco-Decipher when Glyco-Decipher was compared with other tools (the same settings as other tools). Up to five core structure candidate results were retained to generate the in silico deglycosylated spectrum for each origin spectrum. When inspecting the Y1 intensity in the glycopeptide spectrum, only one core structure was retained to generate the glycan fragment ion removal spectrum if the relative intensity of matched Y1 ion was more than 0.9. The newly generated.mgf file was searched with MS-GF+ with the following parameters: peptide mass tolerance: 0.02 Da; enzyme: full trypsin digestion with three maximum missed cleavages; peptide length: 6–40 with mass range: 600–4500; carbamidomethylation at C was set as fixed modification, and oxidation at M was set as variable modification; and isotopic error was set to be 0 to evaluate the accuracy of peptide mass deviation. The peptide identification result was filtered by the $q$ value, and only PSMs with $q$ value < 0.01 were retained.

To evaluate the in silico deglycosylation performance of Glyco-Decipher, we compared it with another glycopeptide identification software, MAGIC[25]. MAGIC (V.1.0.2) was downloaded from http://ms.iis.sinica.edu.tw/COmics/ Software_MAGIC.html, and the oxonium ion list in Supplementary Table 2 was uploaded to overwrite the default value (NeuGc/NeuGc-H$_2$O ions were not included in MAGIC). In this comparison, Glyco-Decipher only output only one deglycosylated spectrum corresponding to the top-scored core structure. The in silico deglycosylation.mgf files generated by MAGIC and Glyco-Decipher were searched with the parameters listed in the Y1 intensity analysis section. HexNAc on N was also considered as a variable modification when HexNAc linked to peptide fragmentation ions was considered. In the identification results from MAGIC spectrum file, only PSM with the least $q$ value was retained if more than one PSM were generated from the original glycopeptide spectrum.

To compare the ability of glycan ion removal between MAGIC and Glyco-Decipher, the number and intensity percentage of peptide fragment ions in in silico deglycosylated spectra originating from MAGIC and Glyco-Decipher were investigated. Peptide identification results were theoretically fragmented into b/y ions at +1/+2 charge states. Fragment ions with ammonium/water loss and isotopic forms (+1/+2) were also added to the list of peptide theoretical fragment ions. The list of peptide fragment ions was matched to the deglycosylated spectra generated by MAGIC and Glyco-Decipher.

**Analysis of mouse dataset and FDR investigation with yeast dataset**. MSFragger (version: 3.1.1) was downloaded to perform an open search with N-glycoproteomics data. The search parameters were as follows: precursor window: lower mass was set to 0 Da, upper mass was set to 3000 Da or 6000 Da; fragment mass tolerance: 20 ppm; enzyme: full trypsin digestion with three maximum missed cleavages; carbamidomethylation at C was set as fixed modification; and oxidation at M was set as variable modification. In MSFragger, the labile search mode was provided for the N-glycan search. The labile search was off for the traditional open search, and it was set to be on for the N-glycan mode open search. The PSM results of the open search with a delta mass (modification mass) less than 892 Da were filtered out because it was less than the mass of the typical core structure of N-glycans.

MSFragger-Glyco (version: 3.1.1), Byonic (version: 3.11.3), StrucGP (version: 1.0.0) and pGlyco 3.0 (version: build20210615) were downloaded. Glyco-Decipher and the other tools were searched with the following parameters: precursor mass tolerance: 5 ppm; fragment mass tolerance: 20 ppm; enzyme: full trypsin digestion with three maximum missed cleavages; carbamidomethylation at C was set as fixed modification; and oxidation at M was set as variable modification. When searching datasets of mouse tissues, the *Mus Musculus* protein database (containing 25,243 entries) was used. In terms of glycan databases, the built-in database of core and branch structures was used in StrucGP and the GlyTouCan database (1766 glycan compositions) was used in MSFragger-Glyco, Byonic, pGlyco 3.0, and Glyco-Deicpher. In addition, phosphorylation on Hex was provided in pGlyco 3.0 as a monosaccaharide modification and was also searched in pGlyco 3.0.

When searching isotopically labeled yeast datasets, the *Schizosaccharomyces Pombe* protein database (containing 5149 entries) downloaded from UniProt was used. In terms of glycan databases, the built-in database of core and branch structures was used in StrucGP and the GlyTouCan database (1766 glycan compositions) was used in MSFragger-Glyco, Byonic, pGlyco 3.0 and Glyco-Deicpher. In addition, ammonium adduction on Hexose was set as a monosaccharide modification in pGlyco 3.0. pQuant[59] was adopted to find isotopically labeled peak pairs. We used the isotopically labeled method in Liu et al.[9] and the reported number of GPSMs with random missing $^{13}$C/$^{15}$N signals to calculate isotopic-based FDR.

In the results of MSFragger-Glyco, glycan masses were retrieved to the original composition in the glycan database. Among the results provided by Byonic, the cutoff score was set to 300, and the | -logProb| threshold was set to 2 to obtain highly confident identification results. GPSM results of StrucGP were directly used in subsequent analysis since FDR screening was already performed in the software. GPSM results with a total FDR < 0.01 with pGlyco 3.0 were retained for comparison with Glyco-Decipher identifications.

**Analysis of human serum dataset**. Glyco-Decipher, StrucGP, and pGlyco 3.0 were searched with the following parameters: precursor mass tolerance: 5 ppm; fragment mass tolerance: 20 ppm; enzyme: full trypsin digestion with three maximum missed cleavages; carbamidomethylation at C was set as fixed modification; and oxidation at M was set as variable modification. The *Homo Sapiens* protein database (containing 20,417 entries) was used. GPSM results of StrucGP were directly used in subsequent analysis since FDR screening was already performed in the software. GPSM results with a total FDR < 0.01 with pGlyco 3.0 were retained for comparison with Glyco-Decipher identifications.

**Analysis of N-linked glycopeptides for the SARS-CoV-2 spike and ACE2**. The raw files of N-glycopeptides of the SARS-CoV-2 spike and ACE2 were downloaded and converted to mzML format by using ProteoWizard with 32-bit precision and the "1-" peak picking option. For the long chromatography gradient used in the original research, a time window of 4000 s was adopted in spectrum expansion during Glyco-Decipher searching. The search parameters adopted in Glyco-Decipher, StrucGP, and pGlyco 3.0 were the same as those used in the original research: precursor mass tolerance: 5 ppm; fragment mass tolerance: 20 ppm; for data from SARS-CoV-2 spike, enzyme settings including a-LP, chymotrypsin, a combination of trypsin and GluC, and a combination of GluC and AspN; for data from ACE2, enzyme settings including a-LP, chymotrypsin, and a combination of trypsin and LysC; cleavage was set to be semispecific; carbamidomethylation at C was set as a fixed modification; and oxidation at M was set as a variable modification. The identification results of pGlyco 2.0 were directly downloaded from PRIDE (PXD019937). GPSM results of StrucGP were directly used in subsequent analysis since FDR screening was already performed in the software. GPSM results with a total FDR < 0.01 with pGlyco 2.0/3.0 were retained for comparison with Glyco-Decipher identifications.

**Implementation, statistics, and visualization**. Glyco-Decipher was implemented in Java (version 11.0.11, https://www.oracle.com/java/). Post-analysis statistics was conducted using Java (version 11.0.11) and Python (version 3.8.3, Anaconda distribution version 4.10.3, https://www.anaconda.com/). The python package "matplotlib" (version 3.4.2), "seaborn" (version 0.11.2) and the JavaScript (version ES2018, https://www.javascript.com/) package "d3" (version 4.13.0) were used for data visualization.

**Reporting summary**. Further information on research design is available in the Nature Research Reporting Summary linked to this article.

## Data availability

Glycoproteomics raw data were downloaded from the PRIDE Archive[54] with accession number PXD005411, PXD005413, PXD005412, PXD005553, PXD005555 (Liu et al.[9] mouse tissues, including the brain, heart, kidney, liver, and lung), PXD005565 (Liu et al.[9] $^{13}$C/$^{15}$N metabolically labeled yeast) and PXD019937 (Zhao et al.[42] SARS-CoV-2 spike and ACE2). Search results (raw data, output files of Glyco-Decipher, StrucGP, pGlyco 3.0, MSFragger-Glyco, and Byonic) that support the findings of this study are available in PRIDE with accession number PXD031032. The mass spectrometry data of human serum have been deposited to the PRIDE repository with accession number PXD031025. Swiss-Prot protein databases used in this study have also been deposited to the PRIDE

Archive and the GlyTouCan database used in this study is provided in Supplementary Data 1. Source data are provided with this paper.

## Code availability

Glyco-Decipher was developed in the Java language and the standalone software package[60] can be downloaded at https://github.com/DICP-1809/Glyco-Decipher.

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

## Acknowledgements

This work was supported, in part, by funds from the National Key Research and Development Program of China (2020YFE0202200 [M.Y.], 2017YFA0505004 [H.Q.]), the National Natural Science Foundation of China (22034007 [M.Y.], 92153302 [M.Y.], 81730016 [Y.N.] and 21775146 [H.Q.]), the LiaoNing Revitalization Talents Program [M.Y.], the innovation program of science and research from the DICP, CAS (DICP & QIBEBT UN201802 [M.Y.], DICP I201935 [M.Y.], DICP I201919 [H.Q.]), Youth Innovation Promotion Association of CAS (2018212 [H.Q.]) and supported by Innovation Academy for Precision Measurement Science and Technology, CAS [M.Y.].

## Author contributions

M.Y. and H.Q. conceived the project. Z.F. developed the algorithm, wrote the software, and analyzed the results. M.D. assisted in the interpretation of the results. J.M. assisted with the algorithm development and software design. Y.N. collected the sample of human serum. Under the supervision of M.D., Z.W. prepared the sample and performed mass spectrometry analyses with MS parameter optimizations. N.Z., Y.W., and L.L. contributed to software development and data analysis. H.Q., M.D. and M.Y. supervised the project. Z.F., H.Q., M.D., and M.Y. discussed and wrote the manuscript.

## Competing interests

The authors declare no competing interests.
