## [Peer Review File · Nature Communications]

Editorial Note: Parts of this peer review file have been redacted as indicated to maintain the confidentiality of other journals.

REVIEWER COMMENTS

Reviewer #1 (Remarks to the Author):

This is a revised manuscript previously submitted to [redacted] and has been reviewed by 3 reviewers. In this revision, the authors have largely answered to all the reviewers' comments and concerns although some not entirely satisfactorily. It packs in a lot of Supplementary notes and Figures, with lots of comparative analysis using other available software. GlycoDecipher is overall a very worthy software tool for glycoproteomics and the authors have invested substantially in this rather well accomplished work, taking into considerations most aspects.

One problem it has is the timing. The main bulk of the work in developing GlycoDecipher and calibrating its performance were obviously completed after MSFragger Glyco came out but before StrucGP and pGlyco3. Therefore the main comparison reported was against Byonic, MSFragger and pGlyco2. In this revision, the authors further added comparison against StrucGP and pGlyco3 as Supplementary data (Suppl Fig 37-38) without actually incorporating the results into the main manuscript other than a casual remark in the Discussion unrelated to their performance. This is unfortunate since pGlyco3 is a significant upgrade from pGlyco2. Given the authors did extend their comparison to pGlyco3 and have the data in hand (and presented as Suppl data), it is more a choice of convenience rather than good scientific rationale to have the main revised manuscript still talking mostly about pGlyco2 instead of replacing it with the data obtained from pGlyco3. This is understandable because doing so would require substantial rewriting and re-doing of the manuscript.

Still, given its find Y1 first and glycan library independent nature, it is best compared against StrucGP and pGlyco3, so that it will not appear much outdated by the time this manuscript is eventually accepted and published. As reported, its performance seems to be significantly better than StrucGP but only marginally so against pGlyco3, which itself is a significant improvement over pGlyco2. The

comparison against StrucGP is also relevant in the sense that both adopted glycan library free approach in the initial find Y1 step but presumably GlycoDecipher outperforms StrucGP by better implementation of the in silico deglycosylation and spectrum expansion steps. Subsequently, it is also interesting to compare their different approaches in putting together the glycan part, both ended up with many unaccounted for "modification"s. GlycoDecipher is seemingly a strong contender among the recently introduced, better performing glycoproteomics/glycopeptide ID software. It is worth adopting and tested by more users, and hence its publication is recommended. Whether it fits well the scope of this journal or better suited for more specialized journal is another matter.

Overall, as already pointed out by previous reviewers, the real novelty and strength of GlycoDecipher is its spectrum expansion step and this is the part that should be highlighted. It was appropriately compared against the open search nature of MSFragger. There are important insights to be learnt here. GlycoDecipher sits somewhere in between pGlyco3 and MSFragger Glyco in terms of its conceptual approach, with comparable or slightly superior performance. The authors stressed too much on the numbers instead of dealing more with the hard questions on how each handles difficult cases. PSM, GPSM and glycopeptide ID numbers are meaningless without considering their reliability. Even the FDR and probability values reported by each software can be misleading when compared side-by-side since the ways these are done for each software are not always directly comparable.

A point to note, as shown by Suppl Table 3, is that a significant gain in additional PSM by spectrum expansion did not actually lead to unique glycopeptide (either unique peptide or unique glycan composition) but simply more of the same glycopeptide from additional MS2 scans of inferior quality. While the increase in total PSM/GPSM is impressive, a more relevant performance index will be how many additional unique glycopeptides were identified with relatively high confidence through this process. GlycoDecipher is seemingly a strong contender among the recently introduced, better performing glycoproteomics/glycopeptide ID software. It is worth adopting and tested by more users, and hence its publication is recommended. Whether it fits well the scope of this journal or better suited for more specialized journal is another matter.

1. Returning to the one strength of GlycoDecipher, it is unclear in the spectrum expansion step whether Y ions are also considered during the search and/or during the similarity scoring. The formula provided for spectrum expansion scoring in the Methods section (In 913) includes m as the number of matched fragment ions of the N-glycan core structure - presumably that means the Y0-Y5 ions? Furthermore it says that "The FDR of the identification results after spectrum expansion was calculated, and PSMs with an FDR<0.01 and with at least 3 core structure fragment ions were retained for further glycan identification.", which means that even those identified by spectrum expansion should at least have 3 Y ions or will be discarded. In other words, please confirm that all GPSM accepted by GlycoDecipher should have at least 3 Y ions - but it needs not be Y0 or Y1? In Suppl Fig 12 legend, it was stated that "Due to the core structure peak validation after spectrum

expansion, all peptide identifications in Glyco-Decipher matched Y1 peaks in MS2". This would imply that presence of Y1 is a criteria for validation. Please clarify the use of Y ions in the search, scoring and subsequent validation.

2. The PSM retained for MS-GF+ search after in silico deglycosylation was based on qValue <0.01. In subsequent spectrum expansion, the scoring of the additional PSM found was based on the formula provided and then controlled by FDR <0.01, with at least 3 core structure fragment ions. What actual PSM scores are these correlated with, respectively, and what values can be considered as confident? Can the PSM scores for both in silico deglycosylation and spectrum expansion be directly comparable? More to the point, as shown in Fig 2d and Suppl Fig 7, the y-axis refers to PSM score and PSM from both in silico deglycosylation and additional spectrum expansion were plotted together. To distinguish from "low score" PSMs and random matches, a PSM score of >100 will seem to be needed. From a user's point of view, once running through the search with GlycoDecipher, which PSM value (score, qValue, FDR, other criteria) should be considered and reported to give a relatively high confident ID that includes PSMs from spectrum expansion? Also, as presented in Suppl Excel Files, each GPSM contains peptide score, core score and glycan score, as well as peptide FDR (but no glycan FDR or expectation value). Are these already filtered at a certain threshold? If so, what's the value?

3. In addition to Suppl Fig 7, which shows the distribution in PSM score for those PSMs identified through spectrum expansion, it will be more convincing to show the actual MS2 spectra for those considered as more convincing (high PSM score) and those among the worst scoring but still acceptable by the FDR threshold. This is important since for all subsequent performance comparison against other search engine, those GPSM identified by spectrum expansion were included.

4. Similarly when trying to match the glycan mass to entries in GlyTouCan, some problematic assignments due to poor monoisotopic pattern, +1 mass offsets, overlapping precursor clusters in chimeric spectra, lack of supportive B and/or Y ions etc should be shown and discussed to let potential users have a better feel of how reliable the ID is, based on the scoring and FDR/probability filtering. All software can handle relatively good quality spectra very well and be able to distinguish even isomeric glycan structures (Suppl Fig 1-3). How each will handle ambiguous case and how these are marked out and reported are actually more relevant than reporting the highly confident cases.

5. Reviewer 1 has correctly pointed out that there are lacdiNAc and fucosylated or sulfated lacdiNAc in the SRAS-CoV2 spike proteins. This is a test of how well GlycoDecipher (and other software) can distinguish the probable terminal structures (and modifications) from mere glycosyl composition. The authors did not respond to the lacdiNAc issue and the +80 modification was attributed to Hex+80 (242 u) on high mannose and not HexNAc+80 or sulfate on complex type N-glycans. Sulfated N-glycan compositions at several sites of the spike proteins were reported by Zhao et al in Suppl Tables S6 and S8.

6. Suppl Fig 1. The data cannot distinguish between what was drawn as a biantennary LeX-LeX versus a triantennary structure.

Reviewer #2 (Remarks to the Author):

I think this can be published with minor corrections.

Lines 53-56: the authors state on average only 6% of glycopeptide spectra are identified whereas more than 70% of spectra can be identified in routine proteomic analysis. I think using these two datasets to represent all of each type of data is somewhat misleading. Matching more than 70% of spectra in a regular proteomics study is unusually high (50% is higher than average), and the last glycopeptide dataset I analyzed in-house I identified 16% of spectra. I think it would be better either to state that only 6% were identified in one of the datasets used in this study whereas a study identifying more than 70% of spectra have been reported for routine proteomic study, or better simply that a significantly lower percentage of spectra are typically identified in glycopeptide datasets.

The authors need to be careful to not imply that if a spectrum does not contain a Y1 then it is incorrect. In the comparison to other software they state that presence of Y1 means the peptide identification is higher confidence, and they state that all the GlycoDecipher results contain a Y1, meaning the GlycoDecipher results are high confidence. The lack of a Y1 does not mean it is incorrect (although it is more likely to be).

Similarly, they need to be careful to not equate 'same answer by both software' as being correct (lines 317-318; just report the agreement).

Line 241: the enlarged mass range in open searching leads to a linear (not exponential) increase in search space: searching 5x larger range makes searches 5x slower.

Figure 4A: I recommend changing 'loss' to 'unique'.

I agree with one of the other reviewers that there should be some comment about what types of data this software may not be as effective with. It is reliant on getting extensive coverage of the smaller Y ions and particularly Y1. These ions will often be present in stepped HCD data. However, these are not as consistently there in data acquired at only one high collision energy, which is more common for data acquired on non-Thermo instruments, and they are also not as regularly present in ETHCD data.

Two reviewers commented that a 30 minute window for matching glycopeptides from the same peptide seems too large. In response they have produced Supplementary Figure 8, which basically confirms this is the case; there is practically no gain going above 20 minutes, and there may not be

any gain going above 15 minutes. I guess the information is now there for a reader to make a decision about whether they would use the same parameters.

The authors have done a lot of data analysis, most of which is in supplementary files/figures. Some of this is of questionable value. For example, there was no need to confirm that the ammonium salt adducts were from an exogenous source: it comes from the buffer (ammonium bicarbonate) used to elute from the HILIC column used for glycopeptide enrichment and is common to most HILIC-enriched glycopeptide datasets. How much superfluous data matters if it is mostly in the supplementary material (there is one sentence in the main manuscript on this), is open to debate.

Reviewer #3 (Remarks to the Author):

The authors have addressed many (but not all) of the raised points, which have improved the manuscript. However, some weaknesses remain. For some of the points, the authors only partially responded to or misunderstood the raised concern. Further, it was unclear from the response letter how the authors actually addressed many of the concerns (they only replied to me, but it was not mentioned how manuscript was actually revised). The following issues need attention:

1. The revised manuscript does not clearly acknowledge the structural uncertainty/ambiguity of the reported glycopeptides and the many shortcomings of the software are not clearly discussed. These points need to be exhaustively addressed to enable readers to understand what this tool can and can't do. For example (but not limited to): i) it needs to be mentioned that the glycan fine structures are not confidently identified with this tool, the top ranked glycan candidates are reported and drawn, but these are not necessarily the correct ones. The structural information is just not available to confidently draw a glycan with fine structural details including topology (e.g. SFig 3), ii) GlcNAc and GlcNAc-Fuc peptides cannot be recognized by this method (many such N-glycopeptides in mouse brain, PMID: 23816992) since no Y2 ions are present for such peptides, iii) Peptide modifications can easily be mistaken as glycan modifications if these are labile PTMs that fall off readily in the HCD process (i.e. the Y1 ion carries the loss of the peptide PTM). The authors did not address these important points in the R1.

2. The authors must also acknowledge that a fair comparison between software may not have been performed with this study - all engines are designed and optimized differently in terms of data input, search settings and output filtering; the authors are experts in using their own tool but not necessarily other's tools. The "superior" performance the authors are reporting should be seen in this light; claiming superior performance in the abstract should therefore be avoided and claim moderated in the rest of the paper. This software is instead better described as another interesting tool that is showing a potential for comprehensive glycoproteomics using a different approach than

other tools. Instead of only focusing on coverage (number of glycoPSMs) informatics papers like this need to also benchmark and “compete” on the accuracy of the IDs (which can only be determined by matching against a ground truth, manually annotated spectra and/or literature, not against output from other search engines as was performed in this work)

3. The authors are still using confusing glycan categories with redundancy in their graphical elements (where does Sia+Fuc glycopeptides fit?, where does paucimannose fit?) It is in my opinion not appropriate to argue that others have grouped glycans like this in previous papers without the appropriate explanation.

4. Glycomics-assisted glycoproteomics was taken out of the revised R1 manuscript despite this being a useful method that is being increasingly used to aid the glycoproteomics analysis; should be reintroduced and cited appropriately.

5. Incorrect Y-ion fragmentation nomenclature still present in the manuscript (as an example, but not limited to, the intact trimannosylchitobiose core is not Y5 since it is not a linear saccharide, see Fig 2 and main text).

1 **Point-to-point response to reviewers' comments**

2 We appreciate the reviewers' insightful and constrictive comments very much to help
3 us improve the manuscript. Additional data analysis and manuscript revision have been
performed to address the reviewers' comments. In this response, the reviewers'
comments are colored in blue and our replies to the comments are colored in black.
Overview of the major changes and updates of the graphical elements of the revisions
are listed at the end of the response.

**The Point-to-Point Response contains:**

Point-to-Point Response for Reviewer#1 Page R3-R52

Point-to-Point Response for Reviewer#2 Page R53-R58

Point-to-Point Response for Reviewer#3 Page R59-R67

Overview of the Major Changes in Manuscript Page R68

Overview of the Updates in Graphical Elements Page R69-R70

Corresponding Relationship of the Figures

Figures in This Response	Figures in the Revised Manuscript
Fig. R1	Fig. 4
Fig. R2	Supplementary Fig. 29
Fig. R3	Supplementary Fig. 28
Fig. R4	Supplementary Fig. 15
Fig. R5	Supplementary Fig. 31
Fig. R6	A figure in the reference PMID: 18429324
Fig. R7	Supplementary Fig. 26
Fig. R8	Supplementary Fig. 7
Fig. R9	Supplementary Fig. 8
Fig. R10	Supplementary Fig. 9
Fig. R11	Supplementary Fig. 42
Fig. R12	Supplementary Fig. 43
Fig. R13a	Supplementary Fig. 33b
Fig. R13b	Supplementary Fig. 33c
Fig. R13c	Supplementary Fig. 33d

**Reviewer #1 (Remarks to the Author):**

**Major**

1. This is a revised manuscript previously submitted to s and has been
reviewed by 3 reviewers. In this revision, the authors have largely answered to all the
reviewers' comments and concerns although some not entirely satisfactorily. It packs in
a lot of Supplementary notes and Figures, with lots of comparative analysis using other
available software. GlycoDecipher is overall a very worthy software tool for
glycoproteomics and the authors have invested substantially in this rather well
accomplished work, taking into considerations most aspects.

**Response:** We thank the reviewer very much for the positive comments and are pleased
to hear that Reviewer #1 found the revisions have improved the original manuscript.

2. One problem it has is the timing. The main bulk of the work in developing
GlycoDecipher and calibrating its performance were obviously completed after
MSFragger Glyco came out but before StrucGP and pGlyco3. Therefore the main
comparison reported was against Byonic, MSFragger and pGlyco2. In this revision, the
authors further added comparison against StrucGP and pGlyco3 as Supplementary data
(Suppl Fig 37-38) without actually incorporating the results into the main manuscript
other than a casual remark in the Discussion unrelated to their performance. This is
unfortunate since pGlyco3 is a significant upgrade from pGlyco2. Given the authors
did extend their comparison to pGlyco3 and have the data in hand (and presented as

Suppl data), it is more a choice of convenience rather than good scientific rationale to
have the main revised manuscript still talking mostly about pGlyco2 instead of
replacing it with the data obtained from pGlyco3. This is understandable because doing
so would require substantial rewriting and re-doing of the manuscript.

**Response:** We thank the positive feedbacks from the reviewer.

In the original manuscript, the performance of Glyco-Decipher was mainly compared
with pGlyco 2.0 since StrucGP and pGlyco 3.0 were not published at the time when the
first submission of the manuscript (StrucGP was published during the first revision of
this manuscript, [reacted] and pGlyco 3.0 was published
during this round of revision, [redacted]). For a better
presentation of the improvements of Glyco-Decipher in glycoproteomics analysis, we
performed comparisons with StrucGP/pGlyco 3.0 instead of pGlyco 2.0 and included
the data in the main text of the latest revised manuscript (Fig. 4). The differences in the
identification strategies between Glyco-Decipher and StrucGP/pGlyco 3.0 were also
discussed in the revised manuscript (Line 516-530).

3. Still, given its find Y1 first and glycan library independent nature, it is best compared
against StrucGP and pGlyco3, so that it will not appear much outdated by the time this
manuscript is eventually accepted and published. As reported, its performance seems
to be significantly better than StrucGP but only marginally so against pGlyco3, which
itself is a significant improvement over pGlyco2. The comparison against StrucGP is

also relevant in the sense that both adopted glycan library free approach in the initial
find Y1 step but presumably GlycoDecipher outperforms StrucGP by better
implementation of the in silico deglycosylation and spectrum expansion steps.
Subsequently, it is also interesting to compare their different approaches in putting
together the glycan part, both ended up with many unaccounted for "modification"s.
GlycoDecipher is seemingly a strong contender among the recently introduced, better
performing glycoproteomics/glycopeptide ID software. It is worth adopting and tested
by more users, and hence its publication is recommended. Whether it fits well the scope
of this journal or better suited for more specialized journal is another matter.

**Response:** We appreciate the constructive and insightful suggestions from the reviewer
and are excited to hear that our work is recommended for publication.

(1) In the revised manuscript, Glyco-Decipher was compared systematically against
StrucGP/pGlyco 3.0 instead of pGlyco 2.0 and the results were included in the main
manuscript (Fig. 4).

(2) StrucGP utilizes the fragment feature of N-glycan core structure rather than the
glycan database in peptide matching, thus also have the potential to identify
glycopeptides with unexpected glycans. Then StrucGP adopts a modularization strategy
for the structural interpretation of site-specific N-glycans by matching B/Y ions in
glycopeptide spectra against a built-in database of glycan core/branch structures. This
strategy allows the discovery of glycans with new/rare structures but fails to cover
glycans with additional modification moiety. The reliance on HCD with different

collision energy to generate good fragmentation for both the peptide and the glycan part,
limited its sensitivity in peptide sequence identification and glycan structure
interpretation, especially for glycopeptides with complex/hybrid glycans (data shown
below).

The rough determination of N-glycan structures, e.g. core fucosylation and bisected
HexNAc, could be achieved by matching diagnostic glycan ions in both StrucGP and
Glyco-Decipher. However, the interpretation of glycopeptide spectra in StrucGP could
only be achieved for those with sufficient glycan ions, which enable the sequencing of
N-Glycans. The modularization strategy in StrucGP also failed to interpret the spectra
of large N-glycans with fragment ions exceed the m/z scan ranges in data acquisition.

As a result, StrucGP only interpreted 77,195 glycopeptide spectra in the dataset of
mouse tissues, which only accounted for 35.9% of the results provided by Glyco-
Decipher (Fig. R1a). Due to the increased complexity of fragmentation of
complex/hybrid N-glycans, StrucGP failed to identify complex/hybrid glycans with
insufficient glycan ions, resulting in only ~1/3 complex/hybrid glycan identifications
compared to the database searching results of Glyco-Decipher/pGlyco 3.0 (Fig. R2b).

These results also indicate that the improvements of Glyco-Decipher over StrucGP are
not only from the better implementation of *in silico* deglycosylation/spectrum
expansion, but also from the more comprehensive coverage in glycan identification.

Despite of the biased identification of glycans in StrucGP, the library-free glycan
identification in StrucGP is of great value for the discovery of unexpected glycan

compositions (rather than modified glycans) beyond database entries. In the
identification results of mouse tissues, StrucGP provided the identification of 227
glycopeptide spectra for 19 unexpected compositions beyond GlyTouCan glycans and
most of them were covered by the glycan profiling of Glyco-Decipher (Fig. R2c). In
contrast, Glyco-Decipher enables a more comprehensive profiling of glycans on
proteins and the implemented monosaccharide stepping also enables the discovery
unexpected glycans and modification moieties on them. In total, Glyco-Decipher
reported as many as 84.0% more glycopeptides in mouse tissues compared to StrucGP
(Fig. R1e).
.

**Figure R1. Comparison between Glyco-Decipher and other software tools.**

This is the same figure as Figure 4 in the revised manuscript.

(a) Comparison of the performance between Glyco-Decipher, StrucGP, pGlyco 3.0, MSFragger-

Glyco and Byonic. Top: Comparison between Glyco-Decipher and other software tools in the

performance of glycopeptide spectrum interpretation. Green: the spectra matched to identical

peptide backbones in Glyco-Decipher and another tool. Orange: the spectra commonly identified in

Glyco-Decipher and another tool but matched to different peptide backbones. Gray: spectra

specifically matched by another tool. Bottom: Comparison between Glyco-Decipher and other

software tools in peptide sequence identification. Green: the peptide sequence commonly identified

by Glyco-Decipher and another tool in pair comparison. Gray: the peptide sequence specifically

matched by another tool in pair comparison.

(b) FDR analysis using the ¹³C/¹⁵N metabolically labeled yeast dataset9. The isotope-based FDR

was calculated by matching ¹³C/¹⁵N isotopic peak pairs in MS1 spectra (line). Red line: FDR

analysis based on the total GPSM results of each tool; blue line: FDR analysis based on the GPSM

results that overlapped with Glyco-Decipher; purple line: FDR analysis based on the GPSMs

specifically identified by each tool. The proportions of GPSMs of oligo-mannose glycans with

composition of Hex(n)HexNAc(2) (green pie), NeuAc (orange pie) or NeuGc (red pie) containing
glycans identified by each tool are shown in the bottom table.

(c) Distributions of GPSMs of ammonium adducted glycans reported by Glyco-Decipher and
pGlyco 3.0.

(d) Number of ammonium adduction GPSMs with/without oligo-mannose glycan composition.

(e) Comparison of glycopeptide identification results between Glyco-Decipher and StrucGP/pGlyco
3.0. The additional (gain), overlap and lost identifications of Glyco-Decipher compared to other
software tools are indicated by blue, green and orange bars, respectively.

(f) Distributions of M6P GPSMs (bars) and intact glycopeptides (dots) across mouse tissues reported
by Glyco-Decipher and pGlyco 3.0. All M6P identifications were validated by the diagnostic
oxonium ion (phosphorylated hexose, $m/z = 243.0269$) in the glycopeptide spectra.

(3) We also replaced the identification results of pGlyco 2.0 by the results of its latest
version: pGlyco 3.0 in the revised manuscript. Compared to pGlyco 2.0, pGlyco 3.0
was updated in many aspects, including glycan search and filtration, and provided
significant improvements in glycopeptide identification. From the dataset of mouse
tissues, a total of 11,082 site-specific glycans were identified by pGlyco 2.0 (data was
shown in the original manuscript) and the number of identified site-specific glycans
was increased to 15,053 in pGlyco 3.0 (Fig. R2a). When the glycan search space was
restricted to the same GlyTouCan database, Glyco-Decipher reported a total of 16,724
site-specific glycans, which provided the increases of 50.9% and 11.1% compared to
the results of pGlyco 2.0 and 3.0, respectively. However, due to the glycan-first search
strategy in pGlyco series tools, the interpretation of glycopeptide spectra is glycan
database-dependent in pGlyco 2.0/3.0 and the absence of specific glycans in the search
space would lead to no assignments of glycopeptides. In contrast, glycan database-
independent peptide search in Glyco-Decipher enables more comprehensive profiling
of glycans in complex samples, and enables the discovery of unexpected glycans

beyond database entries. In addition, the developed monosaccharide stepping method
 also reveals the potential modifications on glycans in discovery mode. Although
 modified glycans could also be searched in pGlyco 3.0 by enumerating database
 glycans with monosaccharide modifications, the search could only be performed with
 user-provided modifications.

**Figure R2. Analysis of the identification results of Glyco-Decipher, StrucGP and pGlyco 3.0**
**on the dataset of mouse tissues.**

This is the same figure as Supplementary Fig. 29 in the revised manuscript.

(a) Identification performance evaluation of Glyco-Decipher in comparison with StrucGP (left) and
pGlyco 3.0 (right) when only identification results of GlyTouCan glycans were considered. The
additional (gain), overlap and lost identifications of Glyco-Decipher compared to other tools are
indicated by blue, green and orange bars, respectively. Compared to StrucGP, 76.5% more
glycopeptide spectra and 58.0% more site-specific glycans were identified by Glyco-Decipher when
the glycan search space was restricted to GlyTouCan database glycans. In comparison with pGlyco
3.0, identical glycan database (GlyTouCan) were adopted in glycopeptide identification, and
increases of 21% and 11.1% in the number of identified glycopeptide spectra and site-specific
glycans were provided by Glyco-Decipher.

(b) Comparison of the glycan identifications in different categories between Glyco-Decipher and
StrucGP/pGlyco 3.0.

(c) Investigation of the glycopeptide spectra of unexpected glycans identified by StrucGP and
pGlyco 3.0.

(d) Comparison of identification performance between Glyco-Decipher and pGlyco 3.0 in the
identification of M6P glycosylation.

We also systematically investigated the performance of Glyco-Decipher and pGlyco

3.0 in the identification of modified glycan on the dataset of yeast: as a ground truth,

only oligo-mannose glycans with composition of Hex(n)HexNAc(2) should be

identified on the proteins of yeast (*Biochimica et Biophysica Acta (BBA) - General*

*Subjects* **1426**, 227-237 (1999)). From the yeast dataset, 7,145 and 5,241 GPSMs were

identified in Glyco-Decipher and pGlyco 3.0, respectively, with the consideration of

ammonium adducted glycans. 4,619 glycopeptide spectra were commonly identified by

Glyco-Decipher and pGlyco 3.0 and 4,590 of them were matched to identical

glycopeptides in the two software tools (Fig. R3a). Among the 29 spectra that were

matched to distinct glycopeptides in Glyco-Decipher and pGlyco 3.0, only 1 of them

was matched to different peptide sequences: the spectra is likely to be a chimeric spectra

since sufficient peptide ions and glycan ions were matched in the spectrum to support
both identifications (as presented in Fig. R3b). For the other 28 spectra, the
identification differences were introduced by non-oligo-mannose glycan identifications
in pGlyco 3.0 while all of them were assigned with oligo-mannose glycans in Glyco-
Decipher. Since the modified glycan identification in pGlyco 3.0 is achieved by
enumerating all modified forms of database entries with provided modification, the
increased glycan search space introduced random matches. For example, as presented
in Fig. R3c, the glycan part was incorrectly matched to an ammonium adducted non-
oligo-mannose glycan in pGlyco 3.0, because the mass difference between Fuc(1)αH(1)
and Hex(2) is 1 Da and the precursor with poor isotopic pattern was assigned to the
(mono-isotope +1 Da) value. In contrast, all the ammonium adducted glycans were
reported to be oligo-mannose glycans by monosaccharide stepping method in Glyco-
Decipher.

From the dataset of yeast, a total of 38 GPSMs of non-oligo-mannose glycans were
reported by pGlyco 3.0 (Fig. R1d). In contrast, all the ammonium adducted glycans
identified by monosaccharide stepping are oligo-mannose glycans, which are agree
with the biosynthesis rule in yeast (Fig. R1d). These results indicate that more
candidates with similar or isotope-shifted masses were introduced in the glycan search
space by the enumeration method implemented in pGlyco 3.0, leading to a decreased
reliability in glycan assignment to some extent. In contrast, the stepwise matching of Y
ions in glycopeptide spectra enables reliable identification of the glycan part

composition and discovers the modification moiety on glycans.
Benefit from the glycan database-independent search and monosaccharide stepping,
164 modified glycans of 27 diverse modification moieties were discovered by Glyco-
Decipher from the dataset of mouse tissues. Combined the results of modified glycans,
Glyco-Decipher provided the identification of 20,651 intact glycopeptides in the mouse
dataset, resulting in a 29.2% increase in glycopeptide identification compared to pGlyco
3.0. Phosphorylation on hexose was also exemplified to demonstrate the performance
of Glyco-Decipher in the identification of modified glycans. 1,044 M6P GPSMs of 157
intact glycopeptides in mouse tissues were identified by the enumeration method in
pGlyco 3.0. In contrast, phosphorylation on hexose were detected in 1,134 glycopeptide
spectra by Glyco-Decipher in discovery mode, leading to the identification of 171 M6P
glycopeptides (Fig. R1f).

In summary, compared to StrucGP and pGlyco 3.0, Glyco-Decipher provides more
comprehensive analysis of glycosylation on proteins. The glycan-independent peptide
searching and monosaccharide stepping also enables the discovery of modified glycans
and reliable elucidation of modification moieties on them.

**Figure R3. Analysis of the identification results of Glyco-Decipher and pGlyco 3.0 on the**
**dataset of yeast.**

This is the same figure as Supplementary Fig. 28 in the revised manuscript.

(a) Analysis of the 4,619 glycopeptide spectra that were commonly identified in Glyco-Decipher
and pGlyco 3.0.
(b) Annotated glycopeptide spectrum of “cwq_mix2-3_726.18611.18611.3” that were matched to
distinct peptide backbones in Glyco-Decipher and pGlyco 3.0.
(c) The MS1 extracted-ion chromatograms (XICs) of the precursor for spectrum “cwq_mix2-2_726.
34572.34572.3” that was differently identified in Glyco-Decipher (left) and pGlyco 3.0 (right). The
peptide and glycan identification results were labeled in the top right of the figure. aH means
ammonium adducted hexose in pGlyco 3.0 and “M” means the mono-isotopic peak of the precursor.
(d) The MS1 extracted-ion chromatograms of the precursor for spectrum “cwq_mix2-
1_726.39623.39623.4” that was matched to the same glycopeptide in Glyco-Decipher (left) and
pGlyco 3.0 (right). The precursor of the spectrum was assigned to the (mono-isotope +1 Da) value
in Glyco-Decipher.

4. Overall, as already pointed out by previous reviewers, the real novelty and strength
of GlycoDecipher is its spectrum expansion step and this is the part that should be
highlighted. It was appropriately compared against the open search nature of
MSFragger. There are important insights to be learnt here. GlycoDecipher sits
somewhere in between pGlyco3 and MSFragger Glyco in terms of its conceptual
approach, with comparable or slightly superior performance. The authors stressed too
much on the numbers instead of dealing more with the hard questions on how each
handles difficult cases. PSM, GPSM and glycopeptide ID numbers are meaningless
without considering their reliability. Even the FDR and probability values reported by
each software can be misleading when compared side-by-side since the ways these are
done for each software are not always directly comparable.

**Response:** We thank the reviewer very much for the constructive suggestions.

(1) Thanks for the suggestion. The more detailed description of the spectrum expansion
algorithm was provided in the Method section of the revised manuscript (Line 640-682).

And the strength of spectrum expansion strategy has also been investigated and
emphasized in the performance evaluation part of the revised manuscript.

(2) We thank Reviewer#1's positive comments on Glyco-Decipher. Both MSFragger-
Glyco and pGlyco 3.0 are glycan database-dependent software tools and could only
interpret glycopeptide spectra with database glycans. Moreover, incorrect peptide
identification might be introduced in glycan-first strategy if inappropriate glycan
databases are adopted. In contrast, the peptide-first and glycan-independent searching
strategy in Glyco-Decipher ensured the comprehensive profiling of glycans in complex
samples. In addition, the utilization of peptide fragmentation feature in spectrum
expansion method also ensures the sensitivity and reliability in peptide identification.

(3) We strongly agree with Reviewer#1's point that the reliability should be the top
priority in glycopeptide identification. In addition to the internal quality control during
data processing, the reliability of Glyco-Decipher has been systematically evaluated in
multiple aspects on different datasets:

(3-1) In the original manuscript, to assess the reliability of peptide identification of
Glyco-Decipher and MSFragger in glycan-independent search, we investigated the
occurrences of Y1 (peptide+HexNAc) ion in the PSMs reported by Glyco-Decipher and
the open search of MSFragger. The results indicated that the corresponding Y1 ions
were matched in all PSMs identified by Glyco-Decipher since corresponding core
structure ion screening, while were matched in only 60-70% PSMs reported by
MSFragger. In the revised manuscript, other Y ions of N-glycan core structure were

also investigated to provide an overview of the distributions of matched core structure
ions for the peptide results (Supplementary Fig. 14 of the revised manuscript). And the
results indicated that the other core structure ions were also matched in higher
percentages of PSMs reported by Glyco-Decipher. To explain this, we investigated the
peptide results of the spectra that were inconsistently identified in Glyco-Decipher and
MSFragger (Fig. R4). Y1 (peptide+HexNAc) ions were matched with higher intensity
in almost all of these spectra based on the peptide results of Glyco-Decipher than that
of MSFragger. Detailed analysis of the identification results from MSFragger indicated
that incorrect peptides with shared peptide fragment ions were matched in open search,
which resulted in the failure in core structure ion matching. A spectrum example was
presented in Fig. R4b: peptide of “VNSTELFHVER” and glycan of
“Hex(9)HexNAc(2)” were matched in this spectrum by Glyco-Decipher and high
intensity corresponding Y ions were matched in this spectrum. Yet the peptide with
additional missed cleavage: “FSVRVNSTELFHVER” was matched by MSFragger for
this glycopeptide spectrum. Note that the peptide of Glyco-Decipher is the C-term
sequence of the peptide result of MSFragger and y1-y10 peptide ions are shared by the
two peptides. However, due to the large MS1 mass window in open search (6,000 Da
for this example), both peptides were included in the peptide candidates. Because
(b/y+HexNAc) ions were also considered in N-glycan mode open search, the peptides
with additional missed cleavage were matched with higher peptide score due to the
interference of glycan ions (labeled in the figure). Accordingly, the glycopeptide

spectrum was matched to the incorrect peptide, and resulted in the lack of
corresponding Y ions of N-glycan core structure in the spectrum. Statistical analysis
indicated that this kind of incorrect matches were commonly observed in the result of
MSFragger for the inconsistently identified spectra (75%-85%, Fig. R4c). Instead of
matching peptide candidates within wide mass windows, the fragment ions of core
structure were matched by Glyco-Decipher to derive the accurate peptide mass, which
largely decreased the search space and ensured reliable peptide identifications.

**Figure R4. Analysis of the glycopeptide spectra that were inconsistently identified in Glyco-**
 **Decipher and MSFragger.**

This is the same figure as Supplementary Fig. 15 in the revised manuscript.

(a) Distributions of intensity differences of the Y1 ion based on the distinct peptide identifications
 in Glyco-Decipher and MSFragger. Intensity of Y1 ion was set to be zero if not matched in the
 spectra. Y1 ions with higher intensity were matched in most of the glycopeptide spectra (>95%)
 based on the peptide results of Glyco-Decipher.

(b) Spectrum example to demonstrate the incorrect peptide matching of MSFragger. The peptide
 identified by Glyco-Decipher, “VNSTELFHVER”, is the C-term sequence part of
 “FSVRVNSTELFHVER”, which was identified by MSFragger. Ten y ions are shared by the two
 sequences (y1-y10) and were matched in both Glyco-Decipher and MSFragger. Yet glycan ions,
 including Y0 ion (peptide) and Y1 ion (peptide+HexNAc), were matched as peptide fragment ions
 incorrectly in MSFragger due to its wider MS1 mass window in open search mode, resulted in the

incorrect peptide identification and the lack of corresponding Y1 ion in the spectrum.
(c) Analysis of the peptide sequence of the spectra that were inconsistently identified by Glyco-
Decipher and MSFragger. In 75-85% of the inconsistently identified glycopeptide spectra, the
peptides of Glyco-Decipher are at the C-term of the peptide results of MSFragger.

(3-2) In the $^{13}\text{C}/^{15}\text{N}$ metabolically labeled yeast dataset, glycan identification results of
99.89% of the GPSMs reported by Glyco-Decipher are oligo-mannose glycans with
compositions of Hex(n)HexNAc(2), which is in line with the biosynthesis rule of
glycosylation on yeast proteins (Fig. R1b). Specifically note that all of the ammonium
adducted glycans are matched to oligo-mannose glycans in Glyco-Decipher, yet 1.8%
of the GPSM results of ammonium adducted glycans were incorrectly matched to non-
oligo-mannose glycans by the enumeration method of pGlyco 3.0 (Fig. R1d), indicating
the high confidence of monosaccharide stepping in modified glycan identification. In
addition, we also calculated the isotope-based FDR by matching isotopic peak pairs in
MS1, the 0.96% isotope-based FDR also suggested the reliability of glycopeptide
identification in the aspect of elemental composition (Fig. R1b).

(3-3) NeuGc-containing N-glycans are unexpected in the glycosylation of human serum
and this feature has been used in a recent community evaluation of glycoproteomics
software tools [redacted]. Based on the analysis of diverse
oxonium/B ions on the acquired human serum dataset, the extracted ion chromatograms
(XICs) indicated that the presence of NeuAc and antenna fucose and the absence of
NeuGc in the glycosylation of human serum (Fig. R5a). No NeuGc-containing glycans
were matched by Glyco-Decipher in the human serum dataset, suggesting the reliability

of Glyco-Decipher in glycan assignment.
 (3-4) As pointed by Reviewer#1, no NeuGc-containing glycans should be identified on
 the protein of SARS-CoV-2 spike, which was cultured in HEK293 T cell. We re-
 analyzed the identification of Glyco-Decipher on the SAR-CoV-2 spike dataset, and
 found that no NeuGc were reported by Glyco-Decipher, which agreed with the
 glycosylation rule (Supplementary Fig. 33a).

 **Figure R5. Analysis of the glycan compositions in human serum.**

This is the same figure as Supplementary Fig. 31 in the revised manuscript.
(a) The XICs of diagnostic oxonium/B ions in the glycopeptide spectra of human serum. The
extracted ion chromatograms were performed at the MS2 level. The XIC results suggested the
presence of NeuAc and antenna fucosylation in the glycopeptides of human serum. In contrast, no
diagnostic oxonium ions of NeuGc were observed in the dataset of human serum. All the mass
spectrometry data with different LC gradients were considered for the XIC analysis and only data
from 210 min LC gradient of were plotted in this figure (similar patterns were observed in data with
different LC gradients and were not shown).
(b) The distributions of site-specific glycans containing NeuGc in the results of Glyco-Decipher,
StrucGP and pGlyco 3.0.
Monosaccharide abbreviation: H: Hex; N: HexNAc; A: NeuAc; G: NeuGc; F: Fuc.

5. A point to note, as shown by Suppl Table 3, is that a significant gain in additional
PSM by spectrum expansion did not actually lead to unique glycopeptide (either unique
peptide or unique glycan composition) but simply more of the same glycopeptide from
additional MS2 scans of inferior quality. While the increase in total PSM/GPSM is
impressive, a more relevant performance index will be how many additional unique
glycopeptides were identified with relatively high confidence through this process.
GlycoDecipher is seemingly a strong contender among the recently introduced, better
performing glycoproteomics/glycopeptide ID software. It is worth adopting and tested
by more users, and hence its publication is recommended. Whether it fits well the scope
of this journal or better suited for more specialized journal is another matter.

**Response:** We thank the positive comments from the reviewer.

(1) Supplementary Table 3 is an example of “MHLNGSNVQVLHR” to illustrate the
principle of spectrum expansion. In Supplementary Table 3, the GPSMs labeled in
green were identified at the *in silico* deglycosylation stage, and the other GPSMs were
the results from spectrum expansion. In this example, four glycopeptides of oligo-

mannose glycans and one glycopeptide of sialylated glycan were identified by *in silico*
deglycosylation. Benefit from spectrum expansion, glycopeptide spectra with inferior
peptide fragmentation were identified along with the unveiling of additional glycan
heterogeneity on the same peptide: additional glycans of Hex(7)HexNAc(2),
Hex(5)HexNAc(4) and a multi-sialylated one Hex(5)HexNAc(4)NeuGc(2) were
uncovered on the peptide backbone after spectrum expansion. To assess the
contribution of spectrum expansion on additional unique glycopeptide identification at
the proteome scale, the identification results of the five mouse tissues were investigated
(glycan search space was restricted to GlyTouCan): in total, 12,924 intact glycopeptides
were initially identified at the initial *in silico* deglycosylation stage; after the
implementation of spectrum expansion, 4,037 additional intact glycopeptides were
uncovered by Glyco-Decipher, leading to an increase of 31.2% compared to the *in silico*
deglycosylation results (Fig. 2e, top).

(2) Dedicated quality control and result evaluation were performed in this work to
ensure reliable glycopeptide identification. During the spectrum expansion process,
PSM score, in which matched peptide ions, their patterns and corresponding core
structure ions were considered, was adopted to evaluate the peptide-spectrum match. In
addition, core structure ions (i.e. Y0,Y-HexNAc(1), Y-HexNAc(2), Y-
Hex(1)HexNAc(2), Y-Hex(2)HexNAc(2) and Y-Hex(3)HexNAc(2)) for the peptide
candidate are matched in the glycopeptide spectrum to avoid incorrect identification of
peptides with shared b/y ions (e.g. peptides with different missed cleavages). Since

different peptides have distinct features, including sequence length, fragmentation
pattern and ion intensity in MS2, the expectation value (e-value) was utilized in FDR
control instead of PSM score: decoy spectra were generated by shifting the m/z values
(1-30) of fragment ions in MS2 randomly and the distribution of PSM scores of target-
decoy spectrum matching are used to calculate the e-value of a PSM during the
expansion process for each peptide (Fig. R6). Then the PSMs are listed in decreasing
order of spectral e-value. The FDR of the PSMs less than a specific e-value is calculated
with the formula: $FDR = (2 * N_{decoy}) / (N_{target} + N_{decoy})$, where N is the number of PSMs
less than the e-value threshold. Then the e-value threshold for PSM identification with
0.01 FDR was derived. And only the PSMs with e-value less than the threshold were
retained for subsequent glycan identification. It is worth note that similar e-value-based
quality control was also adopted in other proteome software tools, e.g. pFind (*Nature*
*Biotechnology* **36**, 1059-1061 (2018)) and MS-GF+ (*Nature Communications* **5**, 5277
(2014)). And we have provided more detailed descriptions of spectrum expansion in
the Method section (Line 640-682).

**Figure R6. Derivation of the expectation value for peptide spectrum matches.**

This is a figure in the reference: *Current Protocols in Protein Science* **49**, 25.22.21-25.22.19 (2007).

(a) Histogram distribution of matching score of PSMs. True-positive match example was indicated
by circle in this figure.

(b) Log-transformed distributions of the matching score of PSMs. Log-transformed survival
distribution of PSM scores was used in this study.

(c) Extrapolation of the linear regression of the data in (b) to high score PSMs. The expectation
value (e-value) is derived from the corresponding y-axis coordinate (the e-value is -8.2 in this
example).

(3) To evaluate the confidence of PSM identifications gained in spectrum expansion,

the following evaluations were performed in this work:

(3-1) We compared the peptide fragmentation patterns of the PSMs originated from *in*

*silico* deglycosylation and the spectrum expansion. Compared to the PSMs initially

identified by *in silico* deglycosylation, the peptide fragmentation pattern stayed

unchanged in the results of spectrum expansion (as presented in Supplementary Fig. 12

of the revised manuscript).

(3-2) In this revision, we reanalyzed the identification results from *in silico*

deglycosylation and spectrum expansion on the mouse dataset (Fig. R7a). Most of the
glycopeptide spectra identified by Glyco-Decipher, either at the *in silico*
deglycosylation stage or at the spectrum expansion stage, were matched to identical
peptide backbones in comparison with other tools, indicating the consistency in peptide
identification between Glyco-Decipher and others (Fig. R7a).

(3-3) We also utilized the $^{13}\text{C}/^{15}\text{N}$ metabolically labeled yeast dataset to assess the
reliability of spectrum expansion. Over 99% GPSMs gained from spectrum expansion
were matched to oligo-mannose glycans in the yeast data (in line with the biosynthesis
rule), and the isotope-based FDR was around 0.92% for the GPSMs additional gained
by spectrum expansion (Fig. R7b).

The above results indicated that high-confidence identification was performed for both
the peptide backbone and the glycan part of the spectrum expansion results.

**Figure R7. Analysis of the identification results from *in silico* deglycosylation (before**
 **expansion) and spectrum expansion.**

This is the same figure as Supplementary Fig. 26 in the revised manuscript.

(a) Comparison of the PSM results achieved by *in silico* deglycosylation (before expansion, left)
 and spectrum expansion (expansion part, right) in Glyco-Decipher with other software tools on the
 dataset of mouse tissues.

(b) FDR analysis of the glycopeptide identification results from *in silico* deglycosylation (before
 expansion, left) and spectrum expansion (right) on the dataset of ¹³C/¹⁵N metabolically labeled yeast
 dataset. The isotope-based FDR was calculated by ¹³C/¹⁵N isotopic peak matching in MS1 with
 pQuant.

**Minor**

1. Returning to the one strength of GlycoDecipher, it is unclear in the spectrum
 expansion step whether Y ions are also considered during the search and/or during the
 similarity scoring. The formula provided for spectrum expansion scoring in the
 Methods section (ln 913) includes m as the number of matched fragment ions of the N-

glycan core structure - presumably that means the Y0-Y5 ions? Furthermore it says that
"The FDR of the identification results after spectrum expansion was calculated, and
PSMs with an FDR<0.01 and with at least 3 core structure fragment ions were retained
for further glycan identification.", which means that even those identified by spectrum
expansion should at least have 3 Y ions or will be discarded. In other words, please
confirm that all GPSM accepted by GlycoDecipher should have at least 3 Y ions - but
it needs not be Y0 or Y1? In Suppl Fig 12 legend, it was stated that "Due to the core
structure peak validation after spectrum expansion, all peptide identifications in Glyco-
Decipher matched Y1 peaks in MS2". This would imply that presence of Y1 is a criteria
for validation. Please clarify the use of Y ions in the search, scoring and subsequent
validation.

**Response:** Thanks. A more detailed description of the use of Y ions during spectrum
expansion has been added in the Method section in the revised manuscript (Line 640-
682).

The use of core structure Y ions in searching, scoring and subsequent filtering is
described as follows:

(1) The use of core structure Y ions in *in silico* deglycosylation stage: The peptide was
initially identified by the *in silico* deglycosylation method, in which the *m/z* gaps of N-
glycan core structure ions (i.e. Y0, Y-HexNAc(1), Y-HexNAc(2), Y-Hex(1)HexNAc(2),
Y-Hex(2)HexNAc(2), Y-Hex(3)HexNAc(2)) were matched in the glycopeptide spectra
for the derivation of the peptide part mass and the subsequent removal of glycopeaks

in MS2.

(2) The use of core structure Y ions in spectrum expansion stage: After the identification
of peptide backbones at the *in silico* deglycosylation stage, the fragmentation pattern of
the peptide backbones were extracted from the initial PSM identification results. Then
the spectrum expansion was performed for each peptide by matching the peptide with
the spectra within the retention time window. For the scoring function in Line 657, the
score to evaluate a peptide-spectrum match (PSM score) contains two part: 1) the
peptide score: this part score evaluates the peptide-spectrum match with matched
peptide ions and their intensity pattern; 2) the core score: in addition to peptide fragment
ions, core structure ions (i.e. Y0, Y-HexNAc(1), Y-HexNAc(2), Y-Hex(1)HexNAc(2),
Y-Hex(2)HexNAc(2), Y-Hex(3)HexNAc(2)) of the peptide were also matched in the
spectrum which aimed to avoid identification of peptides with incorrect mass. And then
the distribution of target PSM scores for this peptide is generated. The decoy spectra
were generated by shifting the m/z value of each fragment ion in target spectra with 1-
30 m/z randomly. And the peptide is also matched to the decoy spectra within the
retention time window to generate the distribution of decoy PSM scores. Since different
peptides have distinct properties, e.g. sequence length, fragmentation pattern and ion
intensities in MS2, the expectation value (e-value) was calculated and used as a
universal threshold in quality control instead of the PSM score. The target-decoy PSM
score distributions are used to calculate the e-value and the e-value threshold for 0.01
spectral FDR was derived during the spectrum expansion process of each peptide. And

the corresponding PSM score threshold for each peptide was also derived accordingly
after e-value filtration. Then the PSM results with less than 3 core structure Y ions (in
which the Y1 ion is required) in their spectra are discarded. The filtration of core
structure ion matching is essential in spectrum expansion to avoid incorrect peptide
identifications with shared b/y ions (e.g. peptides with different number of missed
cleavages but share b or y ions in glycopeptide spectra).

**Reply to each specific question:**

Returning to the one strength of GlycoDecipher, it is unclear in the spectrum expansion step whether
Y ions are also considered during the search and/or during the similarity scoring.

**Reply:** In spectrum expansion, core structure Y ions were matched to calculate the core score and
subsequent filtration for quality control. Only peptide fragment ions were considered in the
similarity scoring.

The formula provided for spectrum expansion scoring in the Methods section (ln 913) includes m
as the number of matched fragment ions of the N-glycan core structure - presumably that means the
Y0-Y5 ions?

**Reply:** Fragment ions of the N-glycan core structure refer to Y0, Y-HexNAc(1), Y-HexNAc(2), Y-
Hex(1)HexNAc(2), Y-Hex(2)HexNAc(2), Y-Hex(3)HexNAc(2).

Furthermore it says that "The FDR of the identification results after spectrum expansion was
calculated, and PSMs with an FDR<0.01 and with at least 3 core structure fragment ions were
retained for further glycan identification.", which means that even those identified by spectrum
expansion should at least have 3 Y ions or will be discarded. In other words, please confirm that all
GPSM accepted by GlycoDecipher should have at least 3 Y ions - but it needs not be Y0 or Y1?

**Reply:** Yes, core structure ion matching was performed after FDR control in spectrum expansion.
PSMs with less than three core structure ions (in which the Y1 (peptide+HexNAc) ion was required)
were discarded. And the statement was added in the revised manuscript (Line 681).

In Suppl Fig 12 legend, it was stated that "Due to the core structure peak validation after spectrum
expansion, all peptide identifications in Glyco-Decipher matched Y1 peaks in MS2". This would
imply that presence of Y1 is a criteria for validation.

**Reply:** Yes, the presence of Y1 ion is a criteria for validation. For a more detailed description, the
sentence in the legend of Supplementary Fig. 12 (which is Supplementary Fig. 13 in the revised
version) has changed to "All PSM identifications were validated by fragment ions of N-glycan core
structure since corresponding ion matching was performed at the identification stages of *in silico*
deglycosylation and spectrum expansion" (Line 369 in revised Supplementary Information).

2. The PSM retained for MS-GF+ search after in silico deglycosylation was based on

qValue <0.01. In subsequent spectrum expansion, the scoring of the additional PSM
found was based on the formula provided and then controlled by FDR <0.01, with at
least 3 core structure fragment ions. What actual PSM scores are these correlated with,
respectively, and what values can be considered as confident? Can the PSM scores for
both in silico deglycosylation and spectrum expansion be directly comparable? More
to the point, as shown in Fig 2d and Suppl Fig 7, the y-axis refers to PSM score and
PSM from both in silico deglycosylation and additional spectrum expansion were
plotted together. To distinguish from "low score" PSMs and random matches, a PSM
score of >100 will seem to be needed. From a user's point of view, once running through
the search with GlycoDecipher, which PSM value (score, qValue, FDR, other criteria)
should be considered and reported to give a relatively high confident ID that includes
PSMs from spectrum expansion? Also, as presented in Suppl Excel Files, each GPSM
contains peptide score, core score and glycan score, as well as peptide FDR (but no
glycan FDR or expectation value). Are these already filtered at a certain threshold? If
so, what's the value?

**Response:** Thanks for these good questions.

(1) MS-GF+ search is only utilized to search peptides from *in silico* deglycosylated
spectra. And the q-value of PSMs from *in silico* deglycosylation, which is the minimum
FDR at which the identification is significant, is also derived based on the score
distributions of PSMs calculated in MS-GF+. Since the scoring function in MS-GF+
only considers the *m/z* and intensity (rather than pattern) information of peptide

fragment ions, the MS-GF+ search only enables the identification of glycopeptide
spectra with sufficient peptide ions. Then we developed spectrum expansion method,
in which the fragmentation pattern of identified peptide backbones was taken into
consideration to improve the glycopeptide identification. The PSM score in this
manuscript indicates the score calculated by the function defined in Line 657, in which
the peptide ions, their intensity pattern and core structure ions are considered. The PSM
score of the PSM results from MS-GF+ search (*in silico* deglycosylation) are re-
calculated with the consideration of peptide ion pattern and core structure ions. The
PSM score is also calculated for PSMs matched in the subsequent spectrum expansion
process.

(2) PSM score is an easy-to-understand assessment for the matching quality of a peptide
with a spectrum and higher PSM score indicates higher confidence in identification.
However, setting fixed PSM score threshold might be inappropriate in the spectrum
expansion process, since different peptides have distinct properties, including sequence
length, fragmentation pattern and ion intensities in MS2. Thus a score threshold for the
expansion of a peptide might be not suitable for another peptide because different
fragmentation patterns are generated in their spectra. Instead of setting PSM score
threshold, we calculated the expectation value (e-value) of each PSM (Fig. R6) and
used this value as a universal assessment for PSMs of different peptides. It is worth
noting that e-value-based quality control has also been used in other proteome tools,
including MS-GF+ (*Nature Communications* **5**, 5277 (2014)) and pFind (*Nature*

*Biotechnology* **36**, 1059-1061 (2018)). The e-value for each PSM is calculated based
on the score distributions of target-decoy PSMs of each peptides. And the target-decoy
PSMs are listed in decreasing order of e-value. The e-value threshold is derived based
on the FDR calculation formula: “ $FDR = (2 * N_{decoy}) / (N_{target} + N_{decoy})$ ”, where N is the
number of PSMs less than an e-value. The quality control in spectrum expansion is
performed by retaining PSMs with e-value less than a threshold for $FDR < 0.01$ at
spectrum value. And the PSM score threshold for each peptide was derived accordingly
after e-value filtration. Then PSMs with $FDR < 0.01$ are considered to be of high
confidence and are retained for subsequent core structure ion matching and glycan
identification. And similar e-value calculation and FDR control are performed in the
identification of the glycan part.

(3) Since distinct scoring algorithms were adopted in Glyco-Decipher and MS-GF+,
their outputted scores are not comparable. In Fig. 2d and Supplementary Fig. 7, MS-
GF+ search enabled the identification of peptide at the first stage (*in silico*
deglycosylation). And the PSM score of these PSMs were re-calculated with the
consideration of peptide fragmentation pattern and were compared with the score
distributions of PSMs that were matched in spectrum expansion. And we also added an
explanation in the figure legend of Fig. 2d: “Specifically note that the score of PSM
from *in silico* deglycosylation was re-calculated with the consideration of peptide
fragmentation pattern and was compared with the score of PSM additional matched in
spectrum expansion”.

(4) When users search glycopeptides with Glyco-Decipher, a folder named
“temp_FileName” is generated under the same directory of the mass spectrometry file,
where “FileName” is the name of the searched mass spectrometry file. The detailed
results generated during glycopeptide identification, including deglycosylated spectra,
PSM results of these spectra and e-value information of the peptide/glycan
identification, are contained in this folder. So users could distinguish whether a PSM is
from spectrum expansion or not and could get more details of the e-value in
peptide/glycan identifications. Since the peptide-first strategy was adopted in Glyco-
Decipher, the PSM results have been filtered at 1% spectral FDR before glycan
identification. And FDR screening has also been performed in Glyco-Decipher after the
identification of glycan part. The results provided in Supplementary excel files have
already been filtered with spectral FDR <1% for both peptide part and glycan part.

**Reply to each specific question:**

What actual PSM scores are these correlated with, respectively, and what values can be considered
as confident?

**Reply:** The PSM score was calculated based on the formula in Line 657, in which the matched
peptide fragment ions, their intensity pattern and the matched core structure ions are considered.
And the PSMs with FDR<0.01 could be considered as high confident results since e-value based
quality control and core structure ion matching were already performed to ensure the identification
reliability.

Can the PSM scores for both in silico deglycosylation and spectrum expansion be directly
comparable? More to the point, as shown in Fig 2d and Suppl Fig 7, the y-axis refers to PSM score
and PSM from both in silico deglycosylation and additional spectrum expansion were plotted
together.

**Reply:** Yes. Although the scores from MS-GF+ search and spectrum expansion are not directly
comparable since the scoring functions are different, in Fig. 2d and Supplementary Fig. 7, the scores
of PSMs from *in silico* deglycosylation (MS-GF+ results) were re-calculated based on the scoring
formula in Line 657, in which the peptide fragmentation pattern and core structure ions were
considered.

To distinguish from "low score" PSMs and random matches, a PSM score of >100 will seem to be

needed.

**Reply:** Instead of setting any fixed PSM score threshold, we calculated the e-value of each PSM
based on the distributions of target-decoy PSMs and utilized the e-value as a universal assessment
for PSMs of different peptides. And the PSM score threshold for each peptide was derived
accordingly after the e-value quality control in spectrum expansion.

From a user's point of view, once running through the search with GlycoDecipher, which PSM value
(score, qValue, FDR, other criteria) should be considered and reported to give a relatively high
confident ID that includes PSMs from spectrum expansion?

**Reply:** During the search process of Glyco-Decipher, a folder named “temp_FileName” is
generated under the same directory of the mass spectrometry file, where “FileName” is the name of
the searched mass spectrometry file. The detailed results generated during glycopeptide
identification, including deglycosylated spectra, PSM results of these spectra and e-value
information of the peptide/glycan identification, are contained in this folder. So users could
distinguish whether a PSM is from *in silico* deglycosylation or spectrum expansion and get more
details of the e-value in peptide/glycan identifications. Among the identification results, the PSMs
with FDR<0.01 could be considered as high confident results since e-value based quality control
and core structure ion matching were already performed to ensure the identification reliability. In
addition,

Also, as presented in Suppl Excel Files, each GPSM contains peptide score, core score and glycan
score, as well as peptide FDR (but no glycan FDR or expectation value). Are these already filtered
at a certain threshold? If so, what's the value?

**Reply:** The results provided in Supplementary excel files have already been filtered with spectral
FDR <1% for both peptide part and glycan part.

3. In addition to Suppl Fig 7, which shows the distribution in PSM score for those PSMs
identified through spectrum expansion, it will be more convincing to show the actual
MS2 spectra for those considered as more convincing (high PSM score) and those
among the worst scoring but still acceptable by the FDR threshold. This is important
since for all subsequent performance comparison against other search engine, those
GPSM identified by spectrum expansion were included.

**Response:** Thanks for the great suggestion. The actual MS2 spectra for the peptides
that were presented in Supplementary Fig. 7a-c of the original manuscript were
provided in Supplementary Fig. 7-9 in the revised manuscript (as presented in Fig. R8-

10 in this response). Specifically, the GPSMs of *in silico* deglycosylation with high
PSM scores were compared against the GPSMs among the worst scoring gained in
spectrum expansion. Their PSM scores, including peptide score and core structure score,
are labeled in the figures. In addition, the fragmentation patterns of peptide backbones
were also compared in detail.

.

**Figure R8. Examples of GPSMs with peptide backbone “THNLTNLPDLQYR” identified**
 **from *in silico* deglycosylation and spectrum expansion.**

This is the same figure as Supplementary Fig. 7 in the revised manuscript.

(a) Score distribution of peptide-spectrum matches in spectrum expansion. Left: score distribution
 of all target-decoy PSMs obtained in spectrum expansion. Right: score distribution of PSMs after
 score filtering and core structure peak matching. And the PSM score threshold was derived from
 the e-value filtration method.

(b) The GPSM initially identified by *in silico* deglycosylation (top) and the GPSM that passed the

score threshold in the spectrum expansion (bottom).
(c) Normalized peaks of peptide part of intact glycopeptides shown in (b). Peptide fragmentation
pattern was retained after spectrum expansion: y4 ion is still the most abundant ion and followed by
y6, y7, b2 ions.

**Figure R9. Examples of GPSMs with peptide backbone “PGDPPGKLDLASGENGTECR”**
 **identified from *in silico* deglycosylation and spectrum expansion.**

This is the same figure as Supplementary Fig. 8 in the revised manuscript.

(a) Score distribution of peptide-spectrum matches in spectrum expansion. Left: score distribution
 of all target-decoy PSMs obtained in spectrum expansion. Right: score distribution of PSMs after
 score filtering and core structure peak matching. And the PSM score threshold was derived from

the e-value filtration method.
(b) The GPSM initially identified by *in silico* deglycosylation (top) and the GPSM that passed the
score threshold in the spectrum expansion (bottom).
(c) Normalized peaks of peptide part of intact glycopeptides shown in (b). Peptide fragmentation
pattern was retained after spectrum expansion: (y11+HexNAc) ion is always the most abundant ion
and followed by (y15+HexNAc) ion.

**Figure R10. Examples of GPSMs with peptide backbone “GGNVTLPCCK” identified from *in***
 ***silico* deglycosylation and spectrum expansion.**

This is the same figure as Supplementary Fig. 9 in the revised manuscript.

(a) Score distribution of peptide-spectrum matches in spectrum expansion. Left: score distribution

of all target-decoy PSMs obtained in spectrum expansion. Right: score distribution of PSMs after
score filtering and core structure peak matching. And the PSM score threshold was derived from
the e-value filtration method.

(b) The GPSM initially identified by *in silico* deglycosylation (top) and the GPSM that passed the
score threshold in the spectrum expansion (bottom).

(c) Normalized peaks of peptide part of intact glycopeptides shown in (b). Peptide fragmentation
pattern was retained after spectrum expansion: y3 ion is the most abundant ion and other peptide
ions are relatively low abundance.

4. Similarly when trying to match the glycan mass to entries in GlyTouCan, some
problematic assignments due to poor monoisotopic pattern, +1 mass offsets,
overlapping precursor clusters in chimeric spectra, lack of supportive B and/or Y ions
etc should be shown and discussed to let potential users have a better feel of how
reliable the ID is, based on the scoring and FDR/probability filtering. All software can
handle relatively good quality spectra very well and be able to distinguish even isomeric
glycan structures (Suppl Fig 1-3). How each will handle ambiguous case and how these
are marked out and reported are actually more relevant than reporting the highly
confident cases.

**Response:** Thanks for the good suggestion. The detailed analysis and discussion on the
precursor detection and glycan identification were added in the revised manuscript:

(1) In the glycosylation of yeast, only oligo-mannose glycans with compositions of
Hex(n)HexNAc(2) are generated on the glycoproteins (*Biochimica et Biophysica Acta*
(*BBA*) - *General Subjects* **1426**, 227-237 (1999)). Based on this knowledge, we
benchmarked the performance of Glyco-Decipher in precursor correction on the yeast
dataset (Fig. R3). Examples of precursor correction for those with poor isotopic pattern

and +1 isotopic error were illustrated in Supplementary Fig. 28c-d in the revised
manuscript (which is the Fig. R3c-d in this response). Incorrect precursor assignments
for 18 GPSMs (0.25% out of 7,145 GPSMs) were observed in the result of Glyco-
Decipher due to the poor MS1 quality. For example, the precursor of glycopeptide
spectrum presented in Fig. R3d was assigned to the (mono-isotope +1 Da) value,
resulted in the identification of oligo-mannose glycan but isotope-shifted modification
part (180 Da for ammonium adducted hexose in this example).

(2) Since the glycosylation on the proteins of mouse are more complex than that of
yeast, chimeric glycopeptide spectra are more likely to be generated in the mouse
dataset. The examples of the identification for the glycopeptide spectra with chimeric
precursors and with insufficient glycan ions in the mouse dataset were discussed in
Supplementary Fig. 42-43 of the revised manuscript (which are presented as Fig. R11-
12 in this response). Among the results of pGlyco 2.0/3.0, 1,145 chimeric glycopeptide
spectra were matched to distinct glycopeptides in pGlyco 2.0 and pGlyco 3.0. Among
the 1,145 glycopeptide spectra, 855 of them were also identified in Glyco-Decipher and
89.4% (764/855) of these spectra were matched to identical glycopeptides as pGlyco
results. For example, in the glycopeptide spectrum in Fig. R11d, peptide backbone of
“ANATIEVK” was matched in all software. Hex(5)HexNAc(6)NeuAc(1)Fuc(1) were
identified as the glycan part in both Glyco-Decipher and pGlyco 2.0 and
Hex(4)HexNAc(6)NeuGc(1)Fuc(2) was reported in pGlyco 3.0. Although identical
mass values are shared by the two glycans, distinct diagnostic oxonium ions were

generated by them and were both detected in this chimeric glycopeptide spectra.

Moreover, for the 91 chimeric glycopeptide spectra that were inconsistently identified
by pGlyco 2.0 and pGlyco 3.0, distinct glycopeptide results were also reported by
Glyco-Decipher. For the spectrum example presented in Fig. R12, identical peptide
backbone was matched by all the three software in the glycopeptide spectrum.
Sufficient peptide fragment ions were matched in the spectrum to support reliable
identification of the peptide backbone, yet only core structure glycan ions were matched
for the identification of glycan part in the three software tools. The identification
differences between the three tools are at the precursor detection and glycan assignment:
In terms of precursor detection, the isotopic pattern and ion intensity in the isolation
window are key factors of precursor correction in Glyco-Decipher. In addition, Glyco-
Decipher also scores glycan candidates of different precursors by matching the
corresponding glycan ions in MS2, and the glycan scoring also helps to determine the
precursor. As a result, the precursor with higher pattern similarity/ion intensity and the
glycan (Hex(6)HexNAc(4)NeuGc(1)Fuc(1)) with more matched glycan ions were
identified in Glyco-Decipher (Fig. R12d, top). The diagnostic oxonium ions of NeuGc
and Y ions of core fucosylation were observed in the spectrum.

Different precursors and glycans were identified in pGlyco 2.0/3.0: Although the
identical precursor for this spectrum was also detected in pGlyco 2.0, another NeuAc-
containing glycan (Hex(7)HexNAc(4)NeuAc(1)) was matched in pGlyco 2.0. Since the
tuned algorithm in precursor detection, a different precursor with less m/z value was

detected in pGlyco 3.0. As described in the original publication [reacted]
pGlyco 3.0 tend to correct the precursor to the one with less m/z
value: “*For potential chimeric spectra, pGlyco3 removes unreliable mixed*
*glycopeptides by determining whether one’s precursor is another’s isotope. For*
*example, if NeuAc(1) and Fuc(2) are simultaneously identified in the same MS2 scan*
*but with different precursors, the Fuc(2)-glycopeptide will be removed because*
*‘NeuAc(1) + 1 Da = Fuc(2)’*”. And the preference was also observed in this spectrum
example: Hex(5)HexNAc(4)NeuGc(2) was identified in pGlyco 3.0 and it corresponds
to the precursor with lowest m/z value but less isotopic similarity and ion intensity in
the isolation window (as shown in Fig. R12b). In this chimeric spectrum example,
distinct glycans were matched in Glyco-Decipher and pGlyco 2.0/3.0. And precursor
and fragment ion evidences are found to support all these glycan identifications. Thus
improvements for precursor detection and glycan assignment are needed for the
glycoproteomics tools for more comprehensive identification of glycopeptides.
.

**Figure R11. Analysis of identification results of Glyco-Decipher and pGlyco 2.0/3.0 on**
 **chimeric glycopeptide spectra.**

This is the same figure as Supplementary Fig. 42 in the revised manuscript.

(a) Analysis of the identification results of 1,145 chimeric glycopeptide spectra that were matched
to distinct glycopeptides in pGlyco 2.0 and pGlyco 3.0. Among the 1,145 glycopeptide spectra, 855
of them were also identified in Glyco-Decipher and the results of Glyco-Decipher were consistent
to pGlyco results in 89.4% (764/855) of these spectra.

(b) Experimental isotopic cluster of the precursor of glycopeptide spectrum “MouseBrain-Z-T-
2.7770.7770.3”. The m/z values of original spectrum precursor and corrected precursors (Glyco-
Decipher, pGlyco 2.0/3.0) were labeled in the figure.

(c) Comparison of the theoretical isotopic pattern with the experimental cluster. The theoretical
isotopic pattern was derived from the elemental composition of identified glycopeptide.

(d) Annotation of the glycopeptide spectra “MouseBrain-Z-T-2.7770.7770.3” based on the
identification results of Glyco-Decipher/pGlyco 2.0 (top) and pGlyco 3.0 (bottom). Diagnostic
oxonium ion and Y ions for glycan identification and structure discrimination (core fucosylation
and bisected HexNAc) were labeled in the figure. From this glycopeptide spectrum, peptide
backbone of “ANATIEVK” was matched in all software. Hex(5)HexNAc(6)NeuAc(1)Fuc(1) were
identified as the glycan part in both Glyco-Decipher and pGlyco 2.0 and
Hex(4)HexNAc(6)NeuGc(1)Fuc(2) was reported in pGlyco 3.0. Although identical mass values are
shared by the two glycans, distinct diagnostic oxonium ions were generated by them and were both
detected in this chimeric glycopeptide spectra.

**Figure R12. Analysis of the chimeric spectrum that was identified to distinct glycopeptides in**
 **Glyco-Decipher and pGlyco 2.0/3.0.**

This is the same figure as Supplementary Fig. 43 in the revised manuscript.

(a) Experimental isotopic cluster of the precursor of glycopeptide spectrum “MouseLung-Z-T-
1.52425.52425.4”. The m/z values of original spectrum precursor and corrected precursor (Glyco-
Decipher, pGlyco 2.0/3.0) were labeled in the figure.
(b) Comparison of the theoretical isotopic pattern with the experimental cluster. The theoretical
isotopic pattern was derived from the elemental composition of identified glycopeptide.
(c) Annotation of the glycopeptide spectrum “MouseLung-Z-T-1.52425.52425.4” with the peptide
backbone “IHLYLQNNFITELPLESFQATGLR” which was commonly identified in Glyco-
Decipher and pGlyco 2.0/3.0.
(d) Annotation of the glycopeptide spectra “MouseLung-Z-T-1.52425.52425.4” based on the glycan
identification results of Glyco-Decipher (top), pGlyco 2.0 (middle) and pGlyco 3.0 (bottom). From
this glycopeptide spectrum, identical peptide backbone was matched in all the three software, and
the identification difference was the glycan part. Hex(6)HexNAc(4)NeuGc(1)Fuc(1) was identified
in Glyco-Decipher and the diagnostic oxonium ions of NeuGc and Y ions of core fucosylation were
matched in the spectrum (top). Hex(7)HexNAc(4)NeuAc(1), which is of the same mass with glycan
reported in Glyco-Decipher, was reported in pGlyco 2.0 and the oxonium ion of NeuAc were also
matched in the spectrum (middle). Hex(5)HexNAc(4)NeuGc(2) was identified in pGlyco 3.0 and it
corresponds to the precursor with lowest m/z value but less isotopic similarity and ion intensity in
the isolation window (as shown in (b)).

5. Reviewer 1 has correctly pointed out that there are lacDiNAc and fucosylated or
sulfated lacDiNAc in the SRAS-CoV2 spike proteins. This is a test of how well
GlycoDecipher (and other software) can distinguish the probable terminal structures
(and modifications) from mere glycosyl composition. The authors did not respond to
the lacDiNAc issue and the +80 modification was attributed to Hex+80 (242 u) on high
mannose and not HexNAc+80 or sulfate on complex type N-glycans. Sulfated N-glycan
compositions at several sites of the spike proteins were reported by Zhao et al in Suppl
Tables S6 and S8.

**Response:** Thanks for the kind reminder from the reviewer.

Glycans with terminal LacDiNAc are significantly produced by HEK293 T cell
(*Molecular Cell* **75**, 394-407.e395 (2019); *Glycobiology*, cwab102 (2021)), which was

used to culture the SAR-CoV-2 Spike protein in the original study (*Cell Host & Microbe*
**28**, 586-601.e586 (2020)). For example, based on the results reported in Zhao et al (*Cell*
*Host & Microbe* **28**, 586-601.e586 (2020)) and prior study of glycosylation on SARS-
CoV-2 spike (*Glycobiology*, cwab102 (2021)), N74 is the site carrying the most amount
of sulfation and additional fucosylation due to the presence of LacDiNAc. The detection
of oxonium ion at m/z 407.17 in the glycopeptide spectra indicated the presence of
LacDiNAc terminal on the glycans identified by Glyco-Decipher (Fig. R13b-c). In
addition, a modification moiety of 283.04 Da, which matches to the mass of sulfated
HexNAc, was identified by monosaccharide stepping method (as presented in Fig.
R13b). The diagnostic ion of HexNAc(1)S(1) at m/z 284.04 in the glycopeptide
spectrum supported the sulfate modification identification on the glycan. Similarly, ions
of HexNAc(1)Fuc(1) at m/z 350.15 and HexNAc(2)Fuc(1) at m/z 553.23 also supported
the presence of fucosylated LacDiNAc terminal (as presented in Fig. R13c), indicating
the performance of Glyco-Decipher in distinguishing terminal structures of glycans.

**Figure R13. Analysis of the modified glycans identified by Glyco-Decipher on the dataset of**
 **SARS-CoV-2 Spike/ACE2.**

(a) The distribution of modified glycans, including glycans linked with moieties of 179 Da (Hex +
 17 Da), 242 Da (Hex + 80 Da) and 283 Da (HexNAc + 80 Da), on the sites of SARS-CoV-2 S (left)
 and ACE2 (right). This is the same figure as Supplementary Fig. 33b in the revised manuscript.

(b) Spectrum example of the glycopeptide with the peptide backbone of “AIHVS⁷⁸GTN⁷⁶GT⁷⁵K” for
 glycosite N74 in SARS-CoV-2 spike and the glycan part of Hex(4)HexNAc(4)NeuAc(1)Fuc(1)
 +283 Da modification moiety identified in Glyco-Decipher. The glycan with sulfated LacDiNAc
 terminal was identified by the monosaccharide stepping method. The diagnostic ions of sulfated
 HexNAc ($m/z = 284.04$) and LacDiNAc ($m/z = 407.17$) were also observed in the glycopeptide
 spectrum. This is the same figure as Supplementary Fig. 33c in the revised manuscript.

(c) Spectrum example of the glycopeptide with the peptide backbone of “AIHVSGTNGTK” for
glycosite N74 in SARS-CoV-2 spike and the glycan part of Hex(3)HexNAc(6)Fuc(2) identified in
Glyco-Decipher. This glycan with fucosylated LacDiNAc terminal was identified by glycan
database searching in Glyco-Decipher. The diagnostic ions of fucosylated HexNAc ($m/z = 284.04$)
and LacDiNAc ($m/z = 407.17$) were also observed in the glycopeptide spectrum. This is the same
figure as Supplementary Fig. 33d in the revised manuscript.

6. Suppl Fig 1. The data cannot distinguish between what was drawn as a biantennary

LeX-LeX versus a triantennary structure.

**Response:** Thank you very much for the kind reminder. Since the determination of fine

glycan structure is still unable to be achieved by Glyco-Decipher, more conservative

illustrations of the structures are provided in the revised manuscript.

**Reviewer #2 (Remarks to the Author):**

I think this can be published with minor corrections.

**Response:** We appreciate the positive feedback from Reviewer#2 and are cheerful to
hear that our work is suitable for publication.

Lines 53-56: the authors state on average only 6% of glycopeptide spectra are identified
whereas more than 70% of spectra can be identified in routine proteomic analysis. I
think using these two datasets to represent all of each type of data is somewhat
misleading. Matching more than 70% of spectra in a regular proteomics study is
unusually high (50% is higher than average), and the last glycopeptide dataset I
analyzed in-house I identified 16% of spectra. I think it would be better either to state
that only 6% were identified in one of the datasets used in this study whereas a study
identifying more than 70% of spectra have been reported for routine proteomic study,
or better simply that a significantly lower percentage of spectra are typically identified
in glycopeptide datasets.

**Response:** Thank you very much for the suggestion. The statement was rephrased to
“For example, in a recently published glycoproteomics dataset of mouse tissues, only
6% of the collected glycopeptide spectra were reliably annotated in the original study
(Supplementary Table 1), a rate that lags far behind routine proteomic analysis, for
which more than 70% of spectrum identification rate has been reported” in the revised
manuscript (Line 54-58).

The authors need to be careful to not imply that if a spectrum does not contain a Y1
then it is incorrect. In the comparison to other software they state that presence of Y1
means the peptide identification is higher confidence, and they state that all the
GlycoDecipher results contain a Y1, meaning the GlycoDecipher results are high
confidence. The lack of a Y1 does not mean it is incorrect (although it is more likely to
be).

**Response:** Thanks for the suggestion.

In the original manuscript, to assess the reliability of peptide identification of Glyco-
Decipher and MSFragger in glycan-independent search, we investigated the
occurrences of Y1 (peptide+HexNAc) ion in the PSMs reported by Glyco-Decipher and
the open search of MSFragger. The results indicated that the corresponding Y1 ions
were matched in all PSMs identified by Glyco-Decipher, while were matched in only
60-70% PSMs reported by MSFragger. In the revised manuscript, other Y ions of N-
glycan core structure were also investigated to provide an overview of the distributions
of matched core structure ions for the peptide results (Supplementary Fig. 14 of the
revised manuscript). And the results indicated that the other core structure ions were
also matched in higher percentages of PSMs reported by Glyco-Decipher. To explain
this, we investigated the peptide results of the spectra that were inconsistently identified
in Glyco-Decipher and MSFragger (Fig. R4). Y1 (peptide+HexNAc) ions were
matched with higher intensity in almost all of these spectra based on the peptide results

of Glyco-Decipher than that of MSFragger. Detailed analysis of the identification
results from MSFragger indicated that incorrect peptides with shared peptide ions were
matched in open search, which resulted in the failure in core structure ion matching. A
spectrum example was presented in Fig. R4b: peptide of “VNSTELFHVER” and
glycan of “Hex(9)HexNAc(2)” were matched in this spectrum by Glyco-Decipher and
high intensity corresponding Y ions were matched in this spectrum. Yet the peptide with
additional missed cleavage: “FSVRVNSTELFHVER” was matched by MSFragger for
this glycopeptide spectrum. Note that the peptide of Glyco-Decipher is the C-term
sequence of the peptide result of MSFragger and y1-y10 peptide ions are shared by the
two peptides. However, due to the large MS1 mass window in open search (6,000 Da
for this example), both peptides were included in the peptide candidates. Because
(b/y+HexNAc) ions were also considered in N-glycan mode open search, the peptides
with additional missed cleavage were matched with higher peptide score due to the
interference of glycan ions (labeled in the figure). Accordingly, the glycopeptide
spectrum was matched to the incorrect peptide, and resulted in the lack of
corresponding Y ions of N-glycan core structure in the spectrum. Statistical analysis
indicated that this kind of incorrect matches were commonly observed in the result of
MSFragger for the inconsistently identified spectra (75%-85%, Fig. R4c). Instead of
matching peptide candidates within wide mass windows, the fragment ions of core
structure were matched by Glyco-Decipher to derive the accurate peptide mass, which
largely decreased the search space and ensured reliable peptide identifications.

Similarly, they need to be careful to not equate ‘same answer by both software’ as being
correct (lines 317-318; just report the agreement.

**Response:** Thanks for the great suggestion.

The high confidence statement based on identical identifications between different tools
were removed in the revised manuscript. And only the consistency in identification was
reported after revision (Line 318-324).

Line 241: the enlarged mass range in open searching leads to a linear (not exponential)
increase in search space: searching 5x larger range makes searches 5x slower.

**Response:** Thanks for the kind suggestion. The sentence was rephrased to “leading to
a substantial increase in the peptide search space” in the revised manuscript (Line 245).

Figure 4A: I recommend changing ‘loss’ to ‘unique’.

**Response:** Thank you very much. The term “loss” has been replaced with “unique” in
the Fig. 4a.

I agree with one of the other reviewers that there should be some comment about what
types of data this software may not be as effective with. It is reliant on getting extensive
coverage of the smaller Y ions and particularly Y1. These ions will often be present in
stepped HCD data. However, these are not as consistently there in data acquired at only

one high collision energy, which is more common for data acquired on non-Thermo
instruments, and they are also not as regularly present in EThcD data.

**Response:** Thanks for the insightful comments on our work.

Due to the high scanning speed of HCD fragmentation and the informative fragment
ions generated with stepped collision energy (SCE), the SCE-HCD fragmentation is of
high sensitivity and is widely used in N-glycopeptide analysis (*Journal of Proteome*
*Research* **19**, 3286-3301 (2020)). Therefore the current version of Glyco-Decipher was
designed for analyzing the data acquired by this kind of fragmentation. Discussions on
the application of our developed tool have been rephrased to: “Glyco-Decipher is
currently designed for the interpretation of N-glycopeptide spectra acquired with
sceHCD fragmentation and supports to accommodate other fragmentation methods and
O-glycosylation analysis will be provided in the near future” in the revised manuscript
(Line 532-535).

Two reviewers commented that a 30 minute window for matching glycopeptides from
the same peptide seems too large. In response they have produced Supplementary
Figure 8, which basically confirms this is the case; there is practically no gain going
above 20 minutes, and there may not be any gain going above 15 minutes. I guess the
information is now there for a reader to make a decision about whether they would use
the same parameters.

**Response:** Thanks for the insightful comments on the spectrum expansion method.

Since a long LC gradient of 360 min was used in acquiring the mouse dataset, thus a
wide expansion window was applied accordingly to avoid the missing of multi-
sialylated glycans on peptide backbones. Indeed, as Reviewer#2 and #3 has pointed out,
search parameters should be selected and optimized properly to get a better
identification performance. And many configuration parameters of Glyco-Decipher, e.g.
expansion window and core structure ion threshold, could be modified by the users to
accommodate the acquired mass spectrometry data.

The authors have done a lot of data analysis, most of which is in supplementary
files/figures. Some of this is of questionable value. For example, there was no need to
confirm that the ammonium salt adducts were from an exogenous source: it comes
from the buffer (ammonium bicarbonate) used to elute from the HILIC column used
for glycopeptide enrichment and is common to most HILIC-enriched glycopeptide
datasets. How much superfluous data matters if it is mostly in the supplementary
material (there is one sentence in the main manuscript on this), is open to debate.

**Response:** Thank you very much for the kind suggestion. Supplementary Note2 has
been removed in the revised manuscript.

**Reviewer #3 (Remarks to the Author):**

The authors have addressed many (but not all) of the raised points, which have
improved the manuscript. However, some weaknesses remain. For some of the points,
the authors only partially responded to or misunderstood the raised concern. Further, it
was unclear from the response letter how the authors actually addressed many of the
concerns (they only replied to me, but it was not mentioned how manuscript was
actually revised). The following issues need attention:

**Response:** We thank the comments from the reviewer to help us improve the
manuscript.

In revision R1, we provided a point-to-point response to address the reviewers'
concerns. In addition, we also provided two word files with changes highlighted for the
main text and the supplementary information, respectively. During this round of
revision, we also appended an overview of the changes in manuscript and figures to
help the reviewers find the improvements of the manuscript.

1. The revised manuscript does not clearly acknowledge the structural
uncertainty/ambiguity of the reported glycopeptides and the many shortcomings of the
software are not clearly discussed. These points need to be exhaustively addressed to
enable readers to understand what this tool can and can't do. For example (but not
limited to): i) it needs to be mentioned that the glycan fine structures are not confidently
identified with this tool, the top ranked glycan candidates are reported and drawn, but

these are not necessarily the correct ones. The structural information is just not available
to confidently draw a glycan with fine structural details including topology (e.g. SFig
3), ii) GlcNAc and GlcNAc-Fuc peptides cannot be recognized by this method (many
such N-glycopeptides in mouse brain, PMID: 23816992) since no Y2 ions are present
for such peptides, iii) Peptide modifications can easily be mistaken as glycan
modifications if these are labile PTMs that fall off readily in the HCD process (i.e. the
Y1 ion carries the loss of the peptide PTM). The authors did not address these important
points in the R1.

**Response:** We thank the useful suggestions from Reviewer#3.

It is important to let users to know the application scope and shortcomings of our
software tool. Thus we have discussed the limitations of Glyco-Decipher in the
Discussion of the revised manuscript (Line 532-543).

(1) During the identification of glycopeptide, the lack of sufficient glycan ions in
glycopeptide spectra limits the determination of glycan structures. In addition, shared
glycan ions are able to be generated by distinct glycan structures, which also hinders
the determination of fine glycan structures. Thus, the determination of fine structure of
glycans is still unable to be achieved in Glyco-Decipher and this limitation was
mentioned in Line 538. The detailed illustrations of glycan structures (e.g. location of
branch structures) were removed in the revised manuscript.

(2) We re-analyzed the data of mouse tissues and found that the identification results of
the two special glycans only accounted for a small percentage of the results: for

example, 24,966 GPSMs were identified in the data of mouse brain by pGlyco 3.0 and
only 101 GPSMs (0.40%) of GlcNAc(1) and 77 GPSMs (0.31%) of GlcNAc(1)Fuc(1)
were reported. However, when the peptide part contains both asparagine and
serine/threonine, the glycosylation of GlcNAc might be incorrectly assigned, especially
for HCD spectra. Thus high confident identification of glycopeptides with the two
special glycans needs further work. In this study, the number threshold in core Y ion
matching was set to be three. To cover glycopeptides of the two special glycans, users
can decrease the number of core Y ions needed in glycopeptide identification (e.g. in
pGlyco 3.0, the number threshold of core Y ions is two), which can be set in the
configuration file of Glyco-Decipher. We also mentioned this in the revised manuscript
(Line 540, Line 624-627) and in the user manual of Glyco-Decipher.

(3) A labile peptide modification might be assigned as a modification on the glycan part
since it was dissociated during the fragmentation of glycan in HCD. However, this type
of modifications might be rare in the glycoproteomics analysis. For example,
phosphorylation is a typical labile PTM and the neutral loss of H₃PO₄ are often
happened in the mass spectra of phosphorylated peptides. In the results of mouse tissues,
phosphorylation on glycans are able to be detected by Glyco-Decipher at high
confidence (FDR<1%) with the validation of diagnostic oxonium ion at *m/z* 243.03.
Moreover, we also mentioned that additional work are needed to get more information
about labile modifications on glycopeptides (Line 542).

2. The authors must also acknowledge that a fair comparison between software may not
have been performed with this study - all engines are designed and optimized differently
in terms of data input, search settings and output filtering; the authors are experts in
using their own tool but not necessarily other's tools. The "superior" performance the
authors are reporting should be seen in this light; claiming superior performance in the
abstract should therefore be avoided and claim moderated in the rest of the paper. This
software is instead better described as another interesting tool that is showing a potential
for comprehensive glycoproteomics using a different approach than other tools. Instead
of only focusing on coverage (number of glycoPSMs) informatics papers like this need
to also benchmark and "compete" on the accuracy of the IDs (which can only be
determined by matching against a ground truth, manually annotated spectra and/or
literature, not against output from other search engines as was performed in this work)

**Response:** Thank you very much for the suggestions.

(1) To ensure the fairness in performance evaluation and comparison, identical search
parameters were set in both Glyco-Decipher and other tools and detailed descriptions
of the parameters were provided in the Methods section (Line 810-875). Identical
protein database, including *Homo Sapiens* (containing 20,417 entries), *Mus Musculus*
(containing 25,243 entries) and *Schizosaccharomyces Pombe* (containing 5,149 entries),
are used in all tools for peptide identification. Trypsin digestion with max missed
cleavage number of 3 were set when analyzing the dataset of mouse, yeast, serum. For
the dataset of SAR-CoV-2 Spike and ACE2, a set of enzymes (trypsin, chymotrypsin,

aLP, AspN and GluC) and their combinations described in the original publication were
set in Glyco-Decipher, StrucGP and pGlyco 3.0. In the identification of glycan, identical
glycan database, which is the GlyTouCan database (containing 1,766 entries), was used
in all tools. Specifically note that StrucGP used the built-in database of core/branch
structures. In post analysis, FDR was set to be 0.01 for Glyco-Decipher and other tools
to obtain high confidence results. In addition, we have uploaded all the raw result files
of Glyco-Decipher and other tools onto the public PRIDE repository, and these results
could be accessed via the link provided in the Data Availability section (Line 876).

(2) It is true that the glycoproteomics software tools are designed and optimized
differently. However, fine-tuning each step in the identification process, e.g. internal
spectrum pre-processing and quality control, is beyond the capability of users in using
of different software tools. Indeed, the search parameter does largely affect the
identification results, and a recently community evaluation also proved this point
[redacted]. Thus, when using other software tools for

comparison, we follow their user manuals and identical parameters (i.e., protein
database, glycan database, enzyme, number of missed-cleavages and mass tolerance
parameters mentioned above) were adopted in glycoproteomics analysis. We have
mentioned importance of search variables in the Introduction section: “Many glycan
database-dependent tools have been developed to analyze the glycoproteomics data
acquired via tandem mass spectrometry (MS/MS) and their performance is closely
associated with the adopted search variables including the glycan database” (Line 42).

(3) Thanks for the suggestions in paper writing. We realized that it is inappropriate to
use “superior performance” when comparing with other software tools. The statements
in Abstract and Main text were moderated and rephrased in the revised manuscript
(Line 27). Beside focusing on the identification performance, we also discussed the
identification strategies in glycopeptide identification and the characteristics of
different glycoproteomics tools (Line 506).

(4) We strongly agree that reliability is the cornerstone of glycopeptide identification.
In addition to paired comparison with other tools, the identification confidence of
Glyco-Decipher was investigated using different datasets by comparison with
knowledge in the literature and manual annotation of glycopeptide spectra:

(4-1) Based on the biosynthesis rule, only oligo-mannose glycans with composition of
Hex(n)HexNAc(2) are modified on the glycosites of yeast proteins (*Biochimica et*
*Biophysica Acta (BBA) - General Subjects* **1426**, 227-237 (1999)). Among the
identification results of Glyco-Decipher, 99.89% of the GPSMs reported by Glyco-
Decipher matched oligo-mannose glycans (Fig. R1b). Note that all the 2,528 GPSMs
of ammonium adducted glycans are matched to oligo-mannose glycans, indicating the
high confidence of Glyco-Decipher in glycan assignment (Fig. R1d).

(4-2) NeuGc-containing glycans are negligible features of glycosylation in human
serum and the diagnostic oxonium ion for this type of glycans was barely detected in
the acquired serum data (Fig. R5a). The absence of NeuGc-containing glycans in
human serum has also been used as a criteria to assess the identification reliability of

glycoproteomics software tools in a recent community study [redacted]
And no glycopeptides of NeuGc-containing glycans were reported in the
results of Glyco-Decipher (Fig. R5b).
(4-3) No NeuGc-containing glycans were matched in the results of Glyco-Decipher
from the SARS-CoV-2 spike data (the SARS-CoV-2 spike proteins were expressed in
the human derived HEK-293 cells), which is also in line with the glycosylation rule
(Supplementary Fig. 33a).
In addition to using the knowledge of glycosylation in reliability evaluation, manual
annotation of the glycopeptide spectra were performed and some spectrum examples
were presented in Supplementary Fig. 1, 2, 3, 6, 7, 8, 9, 15b, 17, 20, 21, 22, 23, 24, 28b,
33c-d, 41c, 42d and 43c-d.
3. The authors are still using confusing glycan categories with redundancy in their
graphical elements (where does Sia+Fuc glycopeptides fit?, where does paucimannose
fit?) It is in my opinion not appropriate to argue that others have grouped glycans like
this in previous papers without the appropriate explanation.
**Response:** Thanks for the comments.
In the original manuscript, we referenced the classification model in a published study
(*Nature Communications* **10**, 1311 (2019)) and classified the glycans into four
categories: high mannose glycans; complex/hybrid glycans (fucose and sialic acid free);
fucosylated glycans (sialic acid free) and sialylated glycans (with/without fucose).

When grouping glycan types, any glycan with sialic acid was categorized as sialylated.
And the fucosylated glycan type group contains any glycan that contains a fucose
moiety and also is not sialylated.
To avoid any possible ambiguity, we classified the identified glycans with a more
universal model in the revised manuscript. The glycans were classified into three non-
redundant categories based on the composition: truncated glycans of
Hex(<4)HexNAc(<3)Fuc(<2); oligo-mannose glycans of Hex(>3)HexNAc(2)Fuc(<2)
and complex/hybrid glycans. To present more composition details of the glycans, the
complex/hybrid glycans were subdivided into four non-redundant subgroups: fucose
and sialic acid free glycans (no fucose and sialic acid in glycan composition);
fucosylated but sialic acid free glycans; sialylated but fucose free glycans and glycans
containing both fucose and sialic acid (Line 766).

4. Glycomics-assisted glycoproteomics was taken out of the revised R1 manuscript
despite this being a useful method that is being increasingly used to aid the
glycoproteomics analysis; should be reintroduced and cited appropriately.

**Response:** Thanks for the suggestion. Glycomics-assisted methods have been cited in
the revised manuscript (Line 538): “The determination of fine glycan structures is
unable to be achieved in Glyco-Decipher and this could be benefited from glycomics
methods, which have been increasingly used in glycosylation analysis¹⁻³”.

References:

1. de Haan, N., Yang, S., Cipollo, J. & Wührer, M. Glycomics studies using sialic acid
derivatization and mass spectrometry. *Nature Reviews Chemistry* **4**, 229-242 (2020).

2. Yamakawa, N. et al. Systems glycomics of adult zebrafish identifies organ-specific
sialylation and glycosylation patterns. *Nature Communications* **9**, 4647 (2018).

3. Tjondro, H.C. et al. Hyper-truncated Asn355- and Asn391-glycans modulate the
activity of neutrophil granule myeloperoxidase. *Journal of Biological Chemistry* **296**,
100144 (2021).

5. Incorrect Y-ion fragmentation nomenclature still present in the manuscript (as an
example, but not limited to, the intact trimannosylchitobiose core is not Y5 since it is
not a linear saccharide, see Fig 2 and main text).

**Response:** Thank you very much for the kind reminder. We checked the labeled Y ions
in all the figures and text of the manuscript. Moreover, the Y ions were denoted with
their definite saccharide compositions (e.g. Y-HexNAc(2) for Y2) in the revised
manuscript.

Overview of the major changes in the revised manuscript:

1. Replaced the comparison with pGlyco 2.0 by the comparisons with StrucGP/pGlyco
3.0.

2. Classify glycans in a more universal model.

3. Added glycopeptide spectrum examples with worst PSM scores in spectrum
expansion to illustrate the utilization of peptide fragmentation pattern in identification.

4. Replaced the detailed glycan structures in figures with more conservative
presentations.

5. Added the analysis of the glycan compositions in human serum.

6. Added examples of precursor with poor isotopic pattern and +1 isotopic error of
Glyco-Decipher.

7. Added discussions on the glycopeptide identification of chimeric spectra.

8. Moderated the performance statements of Glyco-Decipher. Emphasized the
differences between Glyco-Decipher and other tools in glycopeptide identification.

Graphical Element	Description
New Figures	
Supplementary Fig. 7-9	Glycopeptide spectra of the peptide identified from is silico deglycosylation (GPSM with high PSM score) and spectrum expansion (those among worst scoring).
Supplementary Fig. 14	Distributions of matched core structure ions for the peptides identified by Glyco-Decipher and MSFragger.
Supplementary Fig. 15	Analysis of the identification results of glycopeptide spectra that were inconsistently matched in Glyco-Decipher and MSFragger.
Supplementary Fig. 26	Analysis of the results from in silico deglycosylation and spectrum expansion.
Supplementary Fig. 28	Analysis of the differently identified spectra in yeast dataset.
Supplementary Fig. 29	Analysis of the results of Glyco-Decipher, StrucGP and pGlyco 3.0 on the mouse tissues dataset.
Supplementary Fig. 31	Investigation of the glycan compositions in human serum.
Supplementary Fig. 34	The performance comparison between Glyco-Decipher and StrucGP/pGlyco 3.0 on the dataset of SARS-CoV-2 spike and ACE2.
Supplementary Fig. 39	Investigation of the results and relative occupancy rates of site-specific glycans on prosaposin provided by StrucGP.
Supplementary Fig. 40	Investigation of the results and relative occupancy rates of site-specific glycans on prosaposin provided by pGlyco 3.0.
Supplementary Fig. 41-42	Analysis of the identification results of Glyco-Decipher and pGlyco 2.0/3.0 on chimeric spectra in mouse dataset.
Changed Figures	
Fig. 1, 3a, 5	Updated the demonstration of N-glycan structures.
Fig. 2a	Updated the labels of Y ions.
Fig. 2d	Added PSM score threshold derived from e-value method.

Fig. 2e	Updated the classification of glycan types.
Fig. 4	Removed the comparison with pGlyco 2.0. Added the comparison with StrucGP and pGlyco 3.0
Supplementary Fig. 1-3	Updated the demonstration of N-glycan structures.
Supplementary Note 2-4, Fig. 5, 12, 18, 36	Updated the classification of glycan types.
Supplementary Fig. 25	Added the time comparison with StrucGP and pGlyco 3.0. Removed the comparison between GlyTouCan database and the pGlyco 2.0 built-in glycan database.
Supplementary Fig. 30	Updated the comparison with pGlyco 2.0 by the comparisons with StrucGP and pGlyco 3.0 on the human serum dataset.
Supplementary Fig. 33	Added the spectrum examples of glycopeptide with sulfated/fucosylated LacDiNAc terminal in glycan.
Removed Figures/Tables in Original Manuscript	
Supplementary Note 2	Analysis of the ammonium adduction on glycans in yeast dataset.
Supplementary Table 4	List of the 164 modified glycans was provided in Supplementary File 2.
Supplementary Fig. 7	Examples of PSM score distributions of PSMs in spectrum expansion.
Supplementary Fig. 12	Distribution of matched Y1 ions for peptide results of Glyco-Decipher and MSFragger.
Supplementary Fig. 25-28	Protein examples to demonstrate the performance of Glyco-Decipher.
Supplementary Fig. 37	Performance comparison with StrucGP and pGlyco 3.0 on the mouse dataset.
Supplementary Fig. 38	FDR analysis based on the results of Glyco-Decipher and pGlyco 3.0 on the yeast dataset.

REVIEWERS' COMMENTS

Reviewer #1 (Remarks to the Author):

The authors have carefully revised further their manuscript and tried to address all concerns raised by the Reviewers. As commented previously, although Glyco-Decipher is not without its own problems, it does compare favorably with other glycoproteomic software, with some unique aspects that is worth testing out by the community using a wider range of data.

The main strength of a glycan library free spectrum expansion strategy is to ID as many spectra carrying the same peptide backbone and core Y ions but do not have a glycan mass offset (after subtracting the deduced Y0) matched to any entry in the glycan database used, without the need to consider the uncertainty in determining monoisotopic precursors. Moreover, it tolerates all kinds of modifications including some artifacts, and ammonium/cation adducts. As would be expected, a large proportion of candidate glycopeptides and spectra will not be matched as GPSM in Byonic and MSFraggerGlyco due to such issues (i.e. no glycan entries that would add up to the sum of a glycopeptide precursor). On the other hand, Byonic and MSFragger can return PSMs not constrained by presence of Y1, although it is arguable that a large proportion of these will be false positives due to incorrect assignment of peptide modification, mis-cleavages etc. The real test is whether the extra numbers of PSMs and peptides identified by Glyco-Decipher (as opposed to GPSMs) will actually translate into unique glycopeptides with meaningful and reliable glycosyl compositions (and not those with unknown or cation and other modifications of no real significance).

In the Abstract, the authors stated that "Glyco-Decipher provides more comprehensive profiling of glycosylation than other existing tools (Byonic, MSFragger-Glyco, StrucGP and pGlyco 3.0) with at least 34% increases in the number of annotated glycopeptide spectra at high confidence. Perhaps limited by word counts, such statement is misleading. If understood correctly, the "34%" comes from using 215,010 PSM by Glyco-Decipher to compare against the respective numbers from other tools (as presented in Fig 4a). As better described in the Results section (ln 315): "Compared with the listed search tools, Glyco-Decipher reported the most PSMs and provided a 33.5%-178.5% increase in the number of identified glycopeptide spectra due to the glycan database-independent search and spectrum expansion strategy (Fig. 4a, top)."

However, the numbers for MSFraggerGlyco and pGlyco3 actually referred to GPSM matched against entries in GlyTouCan, whereas the PSMs of GlycoDecipher can have any glycan mass including those that did not correspond to database entries and which carry unspecified modifications. A better comparison should be one using GPSM for Glyco-Decipher to compare against others. In fact, one should perhaps compare only the unique glycopeptides identified, without considering all the add-

on modifications, to get a real assessment of the performance. One can then fully acknowledge that additional ID of known and unknown modifications can be considered a true advantage offered by Glyco-Decipher.

In relation to the above, the authors should also define the differences between unique or intact glycopeptides and site-specific glycans.

Another issue to note is that the numbers presented in Fig 2e/Supplemental Fig 29 are different from those presented in Fig 4e. It is 16,971 vs 15,921 unique glycopeptides for Glyco-Decipher and pGlyco3, respectively in the former, but 20,777 and 16,078 intact glycopeptides, respectively in the latter. Although the actual numbers are perhaps irrelevant, it is nonetheless confusing to follow the data and results.

Reviewer #2 (Remarks to the Author):

I am happy with the revised version.

Reviewer #3 (Remarks to the Author):

The authors have addressed the raised concern to my satisfaction. The manuscript appears largely ready for publication. However, before the manuscript can be recommended for publication, the authors should make one minor change in figures and text by changing the use "site occupancy" to "site distribution" or similar wording. Site occupancy is used in the field to describe the site macroheterogeneity (how much of a site is occupied by any glycan, typically as a proportion of 1). Since the non-occupied sites were disregarded in this work due to the reliance of glycopeptide enrichment, the site occupancy information is lost and not considered in the analysis (a fact that would also be appropriate to mention in passing in the manuscript).

**Point-to-point response to reviewers' comments**

We appreciate the reviewers for the great efforts in reviewing the manuscript. We also
thank them very much for the insightful and constructive suggestions to help us improve
the manuscript. In this response, the reviewers' comments are colored in blue and our
replies to the comments are colored in black.

**Reviewer #1 (Remarks to the Author):**

The authors have carefully revised further their manuscript and tried to address all
concerns raised by the Reviewers. As commented previously, although Glyco-Decipher
is not without its own problems, it does compare favorably with other glycoproteomic
software, with some unique aspects that is worth testing out by the community using a
wider range of data.

The main strength of a glycan library free spectrum expansion strategy is to ID as many
spectra carrying the same peptide backbone and core Y ions but do not have a glycan
mass offset (after subtracting the deduced Y0) matched to any entry in the glycan
database used, without the need to consider the uncertainty in determining
monoisotopic precursors. Moreover, it tolerates all kinds of modifications including
some artifacts, and ammonium/cation adducts. As would be expected, a large
proportion of candidate glycopeptides and spectra will not be matched as GPSM in
Byonic and MSFraggerGlyco due to such issues (i.e. no glycan entries that would add
up to the sum of a glycopeptide precursor). On the other hand, Byonic and MSFragger
can return PSMs not constrained by presence of Y1, although it is arguable that a large
proportion of these will be false positives due to incorrect assignment of peptide
modification, mis-cleavages etc. The real test is whether the extra numbers of PSMs
and peptides identified by Glyco-Decipher (as opposed to GPSMs) will actually
translate into unique glycopeptides with meaningful and reliable glycosyl compositions
(and not those with unknown or cation and other modifications of no real significance).

**Response:** We thank Reviewer#1 very much for the positive and insightful comments
on our work.

We agree with Reviewer#1 that the performance should also be tested when only
unmodified glycosyl compositions are considered. In the original manuscript, such
investigations and comparisons, in which only results with GlyTouCan glycans
(without modified glycans) were considered, have been performed on the datasets of
mouse tissues (Supplementary Fig. 29a-b) and human serum (Supplementary Fig. 30c-
34 d). For example, in the dataset of mouse tissues, a total of 215,010 PSMs were identified
by Glyco-Decipher (Fig. 4a) and glycans in 135,840 of them were matched to
GlyTouCan entries (Supplementary Fig. 29a), which are the glycosyl compositions
consisting of Hex, HexNAc, NeuAc, NeuGc and Fuc without any modifications. The
135,840 GPSM results translated to a total of 16,971 unique glycopeptides without the
consideration of modified glycans, and introduced 51% and 7% increases in the number
of glycopeptides compared to the results of StrucGP and pGlyco 3.0, respectively. More
GlyTouCan glycans with complex compositions were identified by Glyco-Decipher
(Supplementary Fig. 29b, Supplementary Fig. 30c-d). In contrast, high false positive
rates of Byonic and MSFragger-Glyco have been reported before (PMID: 24796651;
28874712; 34824474) and high proportional incorrect glycan assignments in Byonic
and MSFragger-Glyco were also observed in this work, which introduced over 10%
FDR in glycopeptide identification (Fig. 4b). These results suggested the improved
performance of Glyco-Decipher in glycopeptide identification even results of modified

glycans were not included.

In the Abstract, the authors stated that "Glyco-Decipher provides more comprehensive
profiling of glycosylation than other existing tools (Byonic, MSFragger-Glyco,
StrucGP and pGlyco 3.0) with at least 34% increases in the number of annotated
glycopeptide spectra at high confidence. Perhaps limited by word counts, such
statement is misleading. If understood correctly, the "34%" comes from using 215,010
PSM by Glyco-Decipher to compare against the respective numbers from other tools
(as presented in Fig 4a). As better described in the Results section (ln 315): "Compared
with the listed search tools, Glyco-Decipher reported the most PSMs and provided a
33.5%-178.5% increase in the number of identified glycopeptide spectra due to the
glycan database-independent search and spectrum expansion strategy (Fig. 4a, top)."
However, the numbers for MSFraggerGlyco and pGlyco3 actually referred to GPSM
matched against entries in GlyTouCan, whereas the PSMs of GlycoDecipher can have
any glycan mass including those that did not correspond to database entries and which
carry unspecified modifications. A better comparison should be one using GPSM for
Glyco-Decipher to compare against others. In fact, one should perhaps compare only
the unique glycopeptides identified, without considering all the add-on modifications,
to get a real assessment of the performance. One can then fully acknowledge that
additional ID of known and unknown modifications can be considered a true advantage
offered by Glyco-Decipher.

**Response:** Thanks for the insightful comments and the constructive suggestion.

(1) We now realized that the original statement in the Abstract might be confusing. It
has been replaced by “Glyco-Decipher reported the most peptide-spectrum matches
than other existing tools (Byonic, MSFragger-Glyco, StrucGP and pGlyco 3.0) with a
33.5%-178.5% increase in the number of identified glycopeptide spectra” (Line 26) for
more accurate description. As pointed by Reviewer#1, one main strength of Glyco-
Decipher is the glycan database-independent spectrum expansion strategy, which
enables the interpretation of glycopeptide spectra that failed to be identified in other
tools. Therefore, the comparison of the glycopeptide spectra that were annotated by
each tool is reasonable. The revised statement is more in line with the features of Glyco-
Decipher and more suitable for the context of the Abstract. Due to word limit, it is hard
to describe the performance of our software in all aspects in the Abstract. And the
systematical evaluations of Glyco-Decipher and comparison with other tools were
described in the main text in various aspects, including search time, FDR analysis, and
comparisons of spectrum/peptide/glycan/glycopeptide results (Fig. 4; Line 307-440;
Supplementary Fig. 25-34).

(2) In the original manuscript, comparisons based on GlyTouCan glycans (without
modified glycans) have been performed on the datasets of mouse tissues
(Supplementary Fig. 29a-b) and human serum (Supplementary Fig. 30c-d). And the
results indicate that Glyco-Decipher exhibits better identification performance even
when the results of modified glycans are excluded. Benefit from glycan database-

independent peptide searching and monosaccharide stepping, glycopeptides with
unexpected glycans are additional gain in glycopeptide identification and provided
deeper insight into micro-heterogeneity of glycosylation.

In relation to the above, the authors should also define the differences between unique
or intact glycopeptides and site-specific glycans.

**Response:** Thanks for the suggestion.

In the original manuscript, the definition of intact glycopeptide in this work was stated:
“Unique (intact) glycopeptides refer to unique peptide backbones and attached glycans
linked on them; that is, glycopeptides with identical peptide backbone but different
glycans refer to different intact glycopeptide identifications” (Line 770 after revision).

In this revision, we added the definition of site-specific glycan in the Method section:
“Unique site-specific glycans refer to unique combinations of glycosites and glycans
linked on them; that is, site-specific glycans with identical glycosite but different
glycans refer to different site-specific glycans.” (Line 772 in the revised manuscript).

Another issue to note is that the numbers presented in Fig 2e/Supplemental Fig 29 are
different from those presented in Fig 4e. It is 16,971 vs 15,921 unique glycopeptides
for Glyco-Decipher and pGlyco3, respectively in the former, but 20,777 and 16,078
intact glycopeptides, respectively in the latter. Although the actual numbers are perhaps
irrelevant, it is nonetheless confusing to follow the data and results.

**Response:** Thanks to Reviewer#1 for the careful review.

We have checked the identification results of Glyco-Decipher and pGlyco 3.0 on the
dataset of mouse tissues, for which the results were presented in Fig. 2e and
Supplementary Fig. 29, and ensured that the numbers of intact glycopeptides are correct.

(1) In Fig. 2e (top) and Supplementary Fig. 29a, the presented results are the
glycopeptides with GlyTouCan glycans, which are 16,971 glycopeptides for Glyco-
Decipher and 15,921 glycopeptides for pGlyco 3.0. And the corresponding information
is mentioned in the original manuscript: In Fig. 2e, “GlyTouCan” was labeled in the x-
axis, and “the glycans in glycopeptide spectra were matched to the downloaded
GlyTouCan database” was mentioned in Line 178 of main text (Line 177 after revision).
In the legend of Supplementary Fig. 29a, we mentioned that “only identification results
of GlyTouCan glycans were considered”.

(2) Benefit from glycan database-independent peptide searching and monosaccharide
stepping, Glyco-Decipher enables the discovery of unexpected modified glycans and
modification moiety on them. In the dataset of mouse tissues, a total of 164 modified
glycans with 27 diverse modification moieties were unveiled by Glyco-Decipher, which
introduced 3,806 glycopeptide identifications with modified glycans ($16,971 + 3,806 =$
$20,777$ glycopeptides for Glyco-Decipher). In pGlyco 3.0, glycopeptides with modified
glycans could only be searched by enumerating user-provided monosaccharide
modification. And Mannose-6-Phosphate (M6P) glycopeptides were searched by
setting phosphorylation on Hex in pGlyco 3.0 to investigate the performance of

modified glycopeptide identification (Fig. 4f) and 157 M6P glycopeptides were
identified ($15,921 + 157 = 16,078$ glycopeptides for pGlyco 3.0). In Fig. 4e, the
presented results contain both glycopeptides with GlyTouCan glycans and those with
modified glycans. We also mentioned this information in the original manuscript:
“Combined with the results of discovered modified glycan” (Line 372, which is Line
373 after revision).

**Reviewer #2 (Remarks to the Author):**

I am happy with the revised version.

**Response:** We appreciate Reviewer#2 for the great efforts in reviewing the manuscript

and are pleased to hear that our revision addressed the raised issues.

**Reviewer #3 (Remarks to the Author):**

The authors have addressed the raised concern to my satisfaction. The manuscript
appears largely ready for publication. However, before the manuscript can be
recommended for publication, the authors should make one minor change in figures
and text by changing the use "site occupancy" to "site distribution" or similar wording.
Site occupancy is used in the field to describe the site macroheterogeneity (how much
of a site is occupied by any glycan, typically as a proportion of 1). Since the non-
occupied sites were disregarded in this work due to the reliance of glycopeptide
enrichment, the site occupancy information is lost and not considered in the analysis (a
fact that would also be appropriate to mention in passing in the manuscript).

**Response:** We are pleased that Reviewer#3 found that our responses addressed the
raised concerns.

We thank the helpful suggestion from Reviewer#3 and we have mentioned that the site
occupancy information was not applicable due to glycopeptide enrichment in the
revised manuscript (Line 760). In addition, we have replaced the term "site occupancy"
with "site-specific glycan distribution" in the revised manuscript (Line 461, and Fig. 5).